# USP32 regulates late endosomal transport and recycling through deubiquitylation of Rab7

Aysegul Sapmaz[1,2,3], Ilana Berlin[1,2,3], Erik Bos[3], Ruud H. Wijdeven[1,2,3], Hans Janssen[1], Rebecca Konietzny[4,6], Jimmy J. Akkermans[2,3], Ayse E. Erson-Bensan [5], Roman I. Koning [3], Benedikt M. Kessler [4], Jacques Neefjes[1,2,3] & Huib Ovaa[1,2,3]

The endosomal system is a highly dynamic multifunctional organelle, whose complexity is regulated in part by reversible ubiquitylation. Despite the wide-ranging influence of ubiquitin in endosomal processes, relatively few enzymes utilizing ubiquitin have been described to control endosome integrity and function. Here we reveal the deubiquitylating enzyme (DUB) ubiquitin-specific protease 32 (USP32) as a powerful player in this context. Loss of USP32 inhibits late endosome (LE) transport and recycling of LE cargos, resulting in dispersion and swelling of the late compartment. Using SILAC-based ubiquitome profiling we identify the small GTPase Rab7—the logistical centerpiece of LE biology—as a substrate of USP32. Mechanistic studies reveal that LE transport effector RILP prefers ubiquitylation-deficient Rab7, while retromer-mediated LE recycling benefits from an intact cycle of Rab7 ubiquitylation. Collectively, our observations suggest that reversible ubiquitylation helps switch Rab7 between its various functions, thereby maintaining global spatiotemporal order in the endosomal system.

[1] Division of Cell Biology, The Netherlands Cancer Institute, Plesmanlaan 121, 1066 CX Amsterdam, Netherlands. [2] Oncode Institute, Department of Cell and Chemical Biology, Leiden University Medical Center, Einthovenweg 20, 2333 ZC Leiden, Netherlands. [3] Department of Cell and Chemical Biology, Leiden University Medical Center, Einthovenweg 20, 2333 ZC  Leiden, Netherlands. [4] Target Discovery Institute, Nuffield Department of Medicine, University of Oxford, Old Road Campus, Headington, Oxford OX3 7FZ, UK. [5] Department of Biological Sciences, Middle East Technical University, Universiteler Mah. Dumlupınar Bulvarı 1, 06800 Çankaya, Ankara, Turkey. [6] Present address: Agilent Technologies, Hewlett-Packard-Strasse 8, 76337 Waldbronn, Germany. These authors contributed equally: Aysegul Sapmaz, Ilana Berlin.  Correspondence and requests for materials should be addressed to J.N. (email: J.J.C. Neefjes@lumc.nl) or to H.O. (email: H.Ovaa@lumc.nl)

The endocytic pathway guards cellular homeostasis through a combination of controlled interactions with the extracellular world and regulated disposal of obsolete or harmful materials[1]. Originating at the cell periphery, this pathway operates via a complex network of progressively maturing carrier vesicles[2]. As early endosomes (EEs) move towards the interior of the cell, they acquire late endosomal (LE) characteristics and become poised to deliver select cargoes for degradation in the lysosome[3]. To protect the endosomal system from the ravages of toxic lysosomal contents, the LE has evolved a gatekeeper function predicated on packaging cargoes destined for degradation into intraluminal vesicles (ILVs). The resulting multi-vesicular body (MVB) serves both as a platform for commitment of cargoes for degradation and as the last point of retrieval[4]. In this way, the MVB constitutes the control center of the endosomal system, with its morphologic and functional integrity bearing directly upon the vesicular network as a whole.

Despite—or perhaps precisely because of—its central position within the endosomal system, cargo and membrane dynamics at the MVB are highly complex, and the manner in which different sorting and trafficking pathways are integrated to best serve its many functions is poorly understood. Over the years, reversible post-translational modification with ubiquitin, orchestrated through the opposition between ligases and deubiquitylating enzymes (DUBs)[5,6], has become recognized as a powerful tool for spatial and temporal control of multi-protein complex assembly[7] central to endosome biogenesis and function[8]. This concept is best illustrated by the profound dependence of endosomal sorting complexes required for transport (ESCRT) on various ubiquitin signals, including ubiquitylation of cargoes as well as ESCRT proteins themselves[9,10]. Cargo sorting to various destinations is further linked to vesicle trafficking carried out by small membrane-associated GTPases. These molecular switches, coupled to discrete vesicular maturation states, direct endosomal transport, fusion, and fission events[11,12], ensuring that this diverse system of vesicles moves and functions in an orderly fashion. Ubiquitylation of several endosomal GTPases has been reported, including EE-bound Rab5[13] and LE/MVB-associated Rab7[14]. Particularly in the case of Rab7—the principal director of membrane traffic to and from proteolytic compartments[15]—the way(s) in which addition and removal of ubiquitylation inform various functions of this GTPase remain obscure. Once Rab7 takes residence on the limiting LE membrane, it can recruit a variety of effector proteins to facilitate diverse processes. These effectors include Rab7-interacting protein (RILP) utilized for anterograde vesicle transport (toward the nucleus)[16] and pleckstrin homology domain-containing family M member 1, along with the associated homotypic fusion and protein sorting complex, for fusion[17,18]. In addition to transport, Rab7 can also direct recycling from the LE membrane to the trans-Golgi network (TGN) and the plasma membrane by cooperating with the retromer complex[19,20]. This begs the question of how Rab7 toggles between anterograde transport and recycling without plunging the MVB into chaos.

In this study, aiming to decipher this conundrum, we consider whether yet undiscovered layers of regulation of ubiquitin dynamics at the MVB membrane influence key decisions in this organelle's biology. Ubiquitylation of Rab7 has recently been shown to promote its association with the retromer and result in extension of tubules from the limiting membrane of the MVB in opposition to ILV formation[14]. Notably, however, no DUB has previously been reported to target Rab7. In a depletion screen for human DUBs affecting surface expression of the LE cargo receptor major histocompatibility class II (MHC-II), we identify USP32 as a powerful regulator of late compartment localization, morphology, and function. Using proteome-wide ubiquitin remnant profiling, we reveal Rab7 to be a key substrate of USP32 and go on to show that USP32 supports Rab7 functions in transport and recycling from the MVB by two different mechanisms. Taken together, our results underscore the nuanced ways in which reversible ubiquitylation can contribute to the ordered complexity of endosomal membrane dynamics.

## Results

**DUB screen for endosomal regulators identifies USP32.** In pursuit of ubiquitin-dependent mechanisms in the regulation of endosomal processes, we performed a small interfering RNA (siRNA)-based screen for DUBs affecting surface levels of MHC-II receptor—a molecule known to traverse the endosomal tract, accumulating in MVBs[21]—in human melanoma MelJuSo cells (Fig. 1a). A number of DUBs previously implicated in endosomal organization and cargo trafficking were picked up with this approach[22–24], including a key endocytic regulator USP8[25,26], depletion of which resulted in lower levels of MHC-II at the cell surface (Fig. 1a). After USP8, the top hit incurring diminished MHC-II surface levels was USP32 (Fig. 1a, Supplementary Fig. 1a, b)—a DUB from the same catalytic family whose cellular function was yet to be described. We hypothesized that DUBs whose loss results in lower receptor surface levels are likely to constitute regulators of endocytic traffic downstream of internalization. To identify these, we further selected our DUB hits on the basis of two intracellular criteria: alterations in (i) distribution and (ii) size of endosomes (Supplementary Fig. 1c, d), taking advantage of the expansive endosomal system in MelJuSo cells, neatly organized into a crowded perinuclear (PN) vesicle cloud and a sparse peripheral contingent[24]. Once again, a striking phenotype was observed with knockdown of USP32 characterized by profound dispersion and swelling of endosomes carrying MHC-II (Fig. 1b–d; Supplementary Fig. 1c, d). Swollen MHC-II endosomes were predominantly late in character, as evidenced by their positivity for the LE marker CD63 (as opposed to the EE marker EEA1), but also contained mannose-6-phosphate (M6PR) receptor (Fig. 1b–d; Supplementary Fig. 1e), which cycles between endosomes and TGN[27,28]. Taken together, altered localization, size, and cargo profile of endosomes affected by the absence of USP32 pointed to broad-spectrum defects in the late compartment.

To test the functionality of the endocytic pathway and its proteolytic competency under suppression of USP32 activity, we examined ligand-mediated trafficking and degradation of the epidermal growth factor (EGF) receptor (EGFR). In HeLa cells treated with control siRNAs, following ligand stimulation, activated EGFR trafficked predominantly to the PN cloud[24], where efficient maturation of its carrier endosomes occurs (Fig. 2a, c). These attributes were disrupted under conditions of USP32 knockdown, as evidenced by dispersion of the endosomal compartment induced by EGF stimulation (Fig. 2a, b) and redistribution of mature EGF-positive structures (i.e., those overlapping with the LE marker CD63) towards the periphery of the cell (Fig. 2a–c). Because proteolytic endosomes and lysosomes are known to largely reside in the PN region[29], we hypothesized that, in the absence of USP32, encounter of activated EGFR with the principal proteolytic enzyme cathepsin D could be hampered. Indeed, a large proportion of EGF-positive endosomes were devoid of cathepsin D in cells depleted of USP32, unlike those in control cells (Fig. 2d, e). Consequently, ligand-mediated degradation of EGFR was strongly inhibited by loss of USP32, leading to prolonged receptor activation (Fig. 2f, g; Supplementary Fig. 2). Taken together, these results implicate USP32 activity in the regulation of the endosomal system's architecture, dynamics, and function.

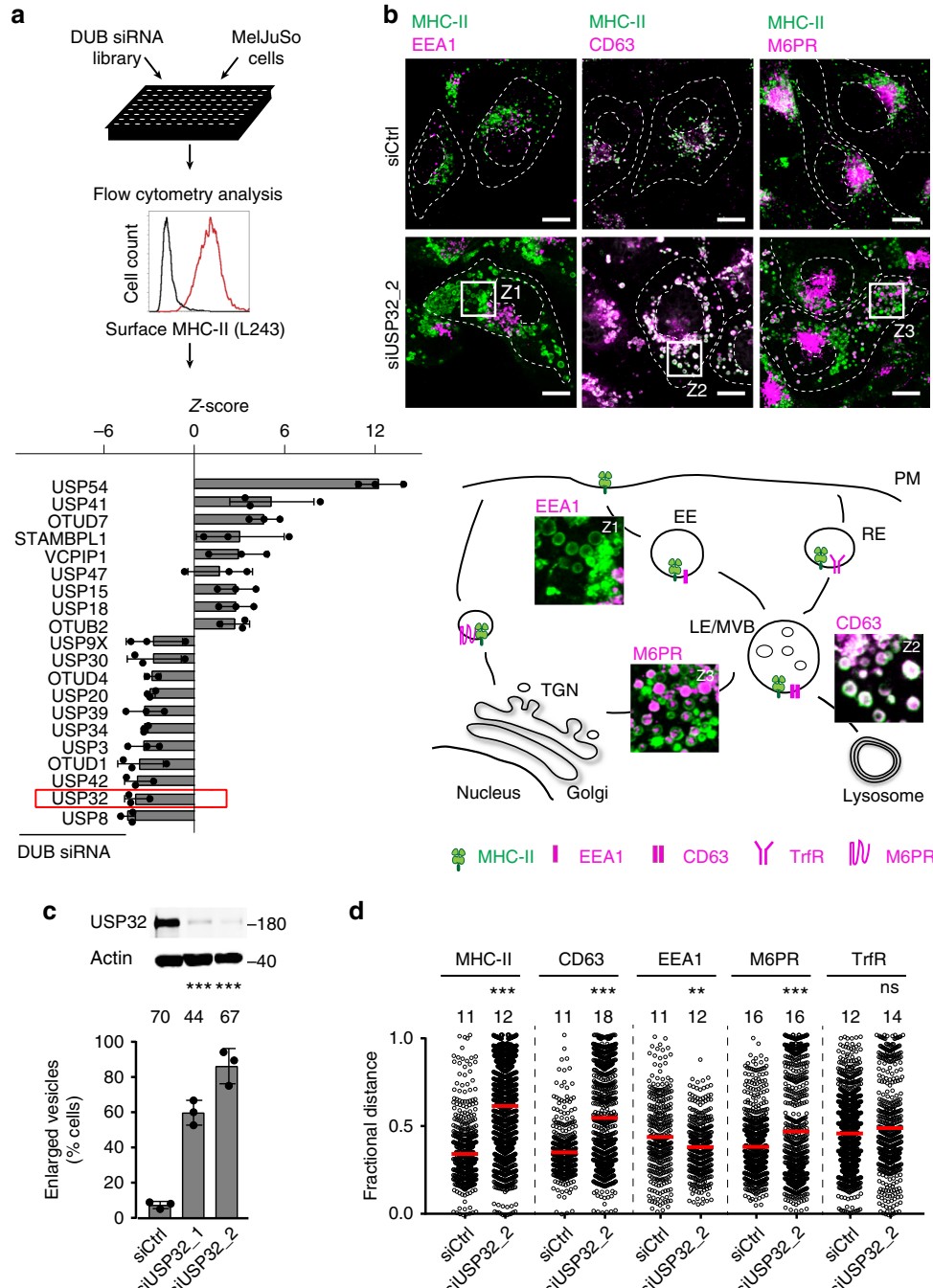

**Fig. 1** Deubiquitylating enzyme (DUB) screen reveals USP32 as a regulator of endosome biology. **a** Small interfering RNA (siRNA)-based screen for DUBs affecting major histocompatability class II receptor (MHC-II) surface levels. MelJuSo cells transfected with siRNAs targeting all human DUBs were analyzed for surface expression of peptide-loaded MHC-II by flow cytometry using monoclonal antibodies (L243-Cy3). Z-scores of DUBs whose depletion resulted in elevated ($Z \geq 3$) or diminished ($Z \leq -3$) levels of MHC-II on the cell surface are plotted, $n = 3$ biologically independent samples. Effect of USP32 depletion is highlighted by a red box. **b** Effect of USP32 loss on the size and distribution of endosomes. Representative confocal overlays of fixed MelJuSo cells transfected with either control (siCtrl) siRNA or oligo #2 targeting USP32 (siUSP32_2) and immunostained against MHC-II (green) and vesicular markers or cargoes (magenta) are shown. EEA1: early endosome (EE) marker, CD63 late endosome (LE)/multi-vesicular body (MVB) marker, mannose-6-phosphate receptor (M6PR): trans-Golgi network (TGN) cargo. Transferrin receptor (TrfR): recycling endosome (RE) marker; PM: plasma membrane. Cell and nuclear boundaries are demarcated with dashed white lines, scale bars = 10 μm. Zooms Z1–Z3 are placed within a schematic of cargo flow. **c** Percentage cells harboring enlarged MHC-II-positive vesicles, $n = 3$ independent experiments. Immunoblot of USP32 protein expression in response to depletion using two independent siRNA oligos (siUSP32_1 and siUSP32_2) is provided with actin as loading control. **d** Vesicle dispersion expressed as fractional distance of MHC-II pixels (black open circles) along a straight line from the center of nucleus (0) to the PM (1.0). Red lines: mean, $n = 2$ independent experiments. Total number of cells analyzed per condition appears above each bar/scatter. Bar graphs report mean of independent sample values (black circles), error bars reflect ±s.d. All significant values were calculated using Student's $t$ test: **$p < 0.01$, ***$p < 0.001$, NS = not significant. See also Supplementary Fig. 1

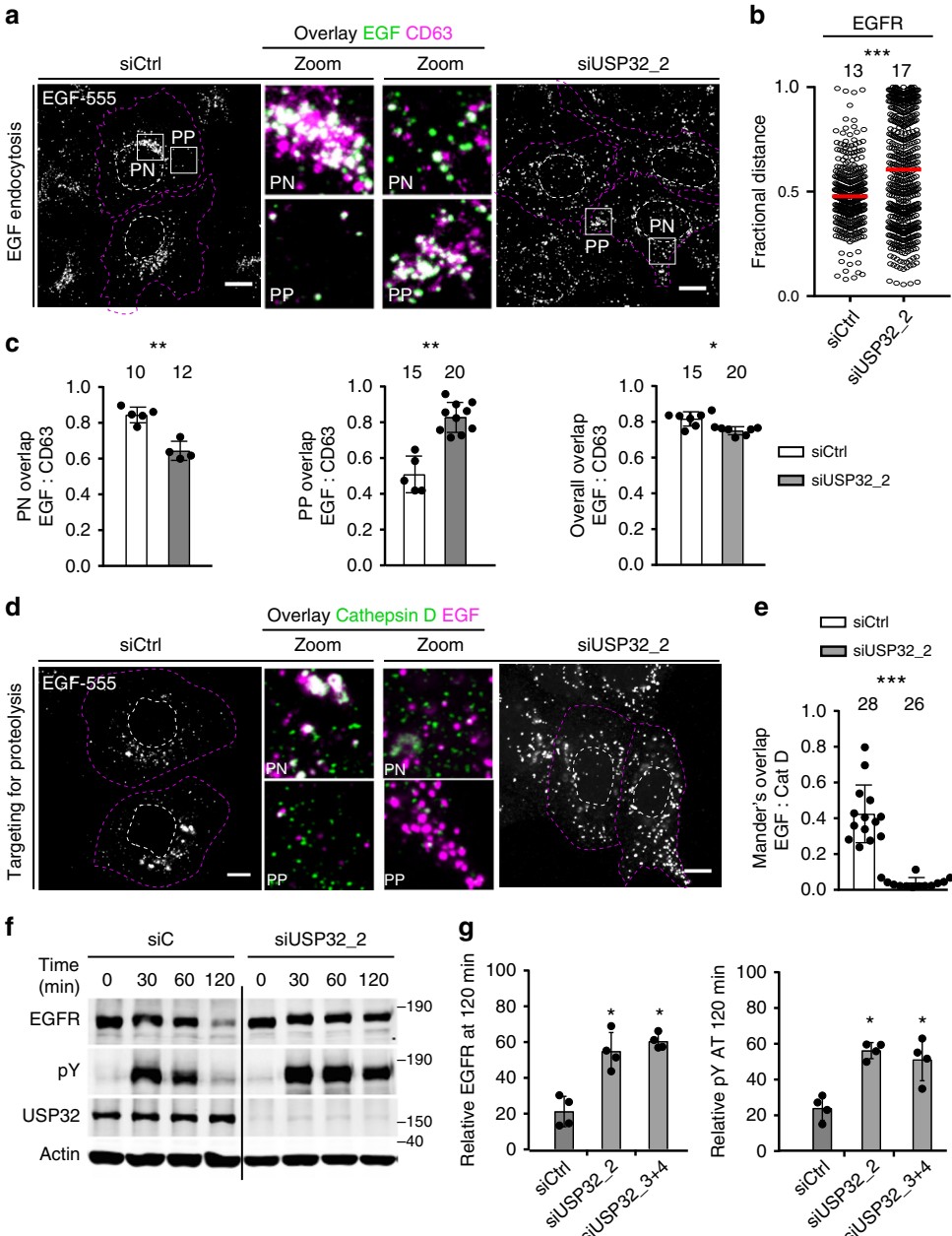

**Fig. 2** Loss of USP32 disrupts cargo trafficking and lysosomal proteolysis. **a–e** Effect of USP32 depletion on ligand-mediated trafficking and degradation of epidermal growth factor (EGF) receptor (EGFR). **a** Representative confocal z-projections of fixed HeLa cells transfected as indicated, starved and stimulated with 100 ng/mL EGF-555 (white) for 120 min. Perinuclear (PN) and peripheral (PP) insets show overlays of EGF (green) with immunostained CD63 (magenta). **b** EGF-positive pixel distribution expressed as fractional distance along a straight line from center of nucleus (0) to the PM (1.0). Red lines: mean, $n = 2$ independent experiments. **c** Colocalization of EGF with CD63 in PN (left), PP (middle), and overall (right) in control cells (siCtrl, white bars) vs. those depleted of USP32 (siUSP32_2, gray bars), $n = 2$ independent experiments. **d** Representative confocal images of fixed HeLa cells transfected as indicated, starved and stimulated with EGF-555 (white) for 120 min. PN and PP insets show overlays of EGF (magenta) with immunostained cathepsin D (green). **e** Colocalization of EGF with cathepsin D in control cells (siCtrl, white bars) vs. those depleted of USP32 (siUSP32_2, gray bars), $n = 3$ independent experiments. All colocalization plots report Mander's overlap quantified from multicell images (black circles). Cell and nuclear boundaries are depicted in dashed magenta and white lines, respectively. Scale bars = 10 μm. **f**, **g** Effect of USP32 depletion on ligand-induced degradation of EGFR. **f** Lysates from HeLa cells transfected as indicated, serum starved, and stimulated with EGF (25 ng/mL) for 0, 30, 60, or 120 min were analyzed by immunoblot against total EGFR (rabbit anti-EGFR) and phosphorylated (pY) EGFR (mouse anti-phosphotyrosine 4G10), with actin as a loading control. **g** Total (left graph, relative to $t = 0$) and activated (right graph, pY relative to $t = 30$) EGFR remaining at 120 min following stimulation in control cells (siCtrl) vs. those depleted of USP32 using different siRNA oligos (siUSP32_2 and siUSP32_3 + 4), $n = 3$ independent experiments. Bar graphs report mean of independent measurements (black circles), error bars reflect ±s.d. Total number of cells analyzed per condition appears above each bar/scatter. All significance calculated using Student's $t$ test: *$p < 0.05$, **$p < 0.01$, and ***$p < 0.001$. See also Supplementary Fig. 2

**USP32 is a membrane-associated catalytically active DUB.** To understand the how USP32 functions in endosome biology, we began by investigating its cellular localization. Endogenous as well as ectopically expressed USP32 was found to associate with the perinuclear TGN and highly dynamic peripheral vesicles (Fig. 3a; Supplementary Fig. 3a, b and Movies 1, 2). We therefore examined the interplay between the LE and TGN as a function of USP32. While in normal cells frequent transient interactions between these compartments were readily observed (Supplementary Movie 3 and 4), loss of USP32 caused TGN-derived membranes to remain stuck on enlarged acidified endosomes (Supplementary Movie 5, 6). Similar findings were observed for depletion of USP8 (Supplementary Movie 7, 8), which has previously been implicated in regulating LE-to-TGN traffic[25]. The vesicular pathway connecting the LE with the TGN is central to the biogenesis of proteolytic organelles and is frequented by the

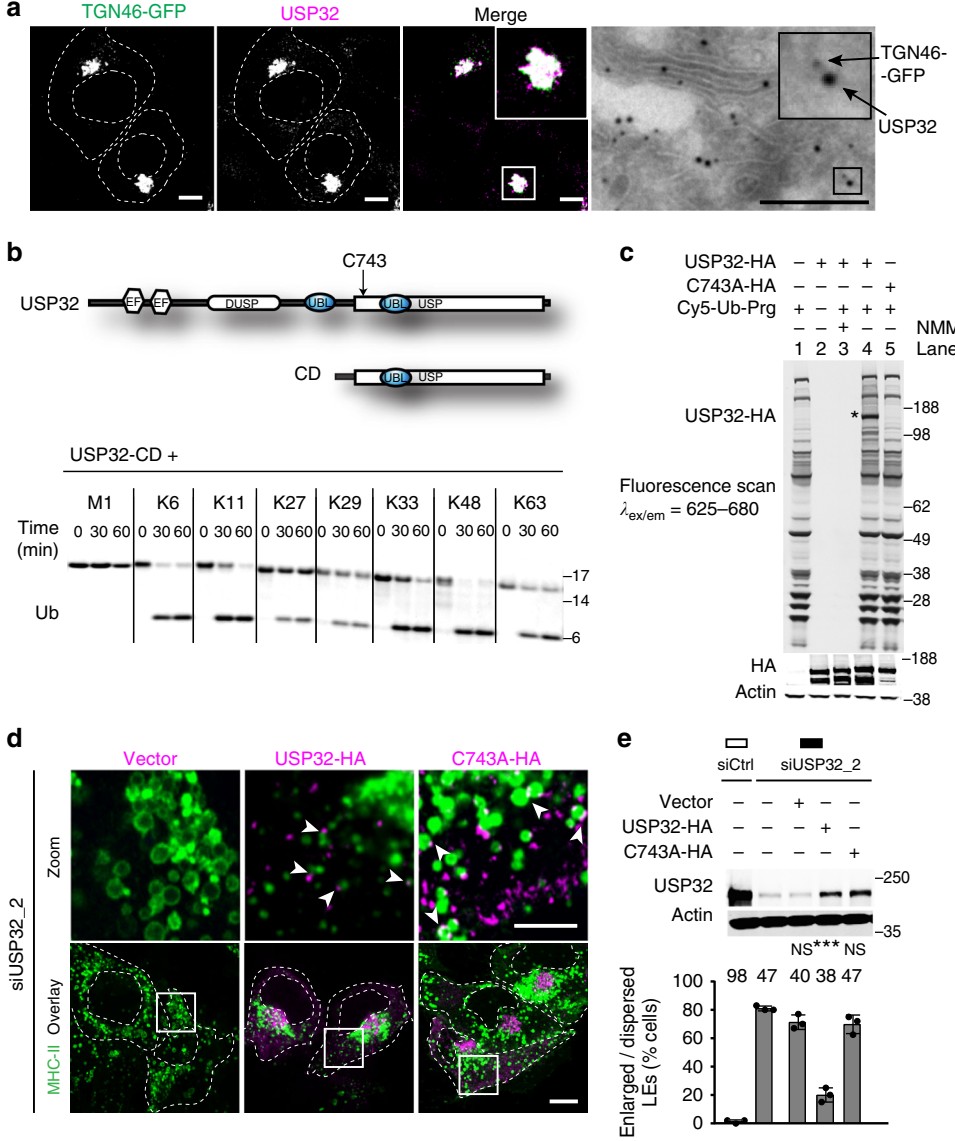

**Fig. 3** Catalytic activity of USP32 supports endosomal system's architecture. **a** Localization of USP32 to the trans-Golgi network (TGN). Left panels: representative confocal images of fixed MelJuSo cells stably expressing TGN46-GFP (green) and immunostained against endogenous USP32 (magenta). Cell and nuclear boundaries are demarcated with dashed white lines, scale bars = 10 μm. Right panels: Electron micrographs of sections co-labeled with anti-USP32 (10 nm gold) and anti-GFP (15 nm gold), scale bar = 0.25 μm. See also Supplementary Fig. 3 and Movies 1–8. **b** Top panel: schematic representation of USP32 domain organization: EF, calcium-binding domain; DUSP, domain found in ubiquitin-specific proteases (USP); UBL, ubiquitin-like domain; USP, catalytic domain harboring the principal catalytic residue C743. Bottom panel: in vitro cleavage of di-ubiquitin linkages (M1, K6, K11, K27, K29, K33, K48, and K63) by the catalytic domain (CD) of USP32. See also Supplementary Fig. 4. **c** DUB activity-based probe assay performed on lysates of HEK293T cells expressing USP32-HA or catalytic mutant C743A-HA in the absence (−) or presence (+) of Cy5-Ub-Prg probe. Reaction products were analyzed using in-gel fluorescence scanning followed by immunoblot against HA; * indicates USP32-HA labeling. **d, e** Rescue of USP32 depletion phenotypes of LE enlargement and dispersion by re-expression of siUSP32_2-resistant USP32-HA vs. C743A-HA relative to empty vector.
**d** Representative confocal images of fixed MelJuSo cells transfected as indicated and immunostained against major histocompatibility class II (MHC-II) (green) and HA (magenta) are shown with the corresponding zooms. Cell and nuclear boundaries are demarcated with dashed white lines. Arrows point to HA-positive puncta juxtaposed to endosomes, scale bars = 10 μm. **e** Enlargement and dispersion of MHC-II-positive vesicles reported as % cells. Bars depict mean of n = 3 independent experiments (black circles), error bars reflect ±s.d., total number of cells analyzed per condition appears above each bar. Immunoblot against USP32 is shown with actin as a loading control. All significance was calculated using Student's t test: ***p < 0.001, NS = not significant

proteolytic enzyme carrier mannose-6-phosphate receptor (M6PR)[30,31]. In control cells pulsed with an antibody recognizing M6PR, efficient trafficking to the PN region was observed during the chase period. By contrast, in cells depleted of USP32, anti-M6PR remained dispersed in vesicular structures (Supplementary Fig. 3c, d), implying a defect consistent with mislocalization of M6PR to enlarged endosomes described in Fig. 1b, d. Taken together with perturbations in trafficking and downregulation of EGFR (Fig. 2), these results position USP32 at a crossroads of multiple vesicular pathways converging at the MVB.

To begin dissecting the nature of USP32 DUB function and its role in endosome biology, we performed biochemical characterization of USP32 catalytic activity. Both the full-length enzyme and its C-terminal fragment harboring the USP domain readily cleaved mono- and di-ubiquitin substrates in vitro (Fig. 3b; Supplementary Fig. 4a, b)[32]. Although USP32 was recently reported to interact preferentially with K6- and K29-linked ubiquitin chains[33], our analysis of all ubiquitin linkage types (M1, K6, K11, K27, K29, K33, K48, and K63) did not reveal striking cleavage preferences by this DUB (Fig. 3b; Supplementary Fig. 4b). As expected, wild-type USP32, but not its catalytic mutant C743A, ectopically expressed in mammalian cells, labeled with a DUB activity-based probe (Fig. 3c), and this catalytic determinant proved necessary to afford rescue of USP32

depletion with respect to both size and localization of endosomes carrying MHC-II (Fig. 3d, e). Unable to rescue loss of the endogenous enzyme, catalytically inactive USP32-C743A localized to discrete patches on enlarged MHC-II vesicles (Fig. 3d, e), suggesting that its activity likely targets endosomal constituents.

**LE GTPase Rab7 is a substrate of USP32.** To identify relevant USP32 substrates, we performed proteome-wide ubiquitome analyses under conditions of varying USP32 abundance. Stable isotope labeling of amino acids in cell culture (SILAC), followed by purification using antibodies recognizing Lys-ε-Gly-Gly remnants of ubiquitylated proteins subjected to trypsin digestion[34], was used to profile and compare ubiquitomes derived from different samples (Fig. 4a). Two complimentary perturbations were tested: (i) depletion and (ii) overexpression of USP32 (performed in MelJuso or HeLa cells, respectively). Given our interest in endosomes, we focused further analysis of the data sets on proteins known to function in this pathway. Interestingly, changes in the ubiquitylation status of the LE master regulator small GTPase RAB7A, henceforth referred to simply as Rab7, were detected in response to altered USP32 abundance. Elevated ubiquitylation of its lysine 191 was observed with USP32 depletion using the SILAC strategy (Fig. 4b; Supplementary Fig. 5a),

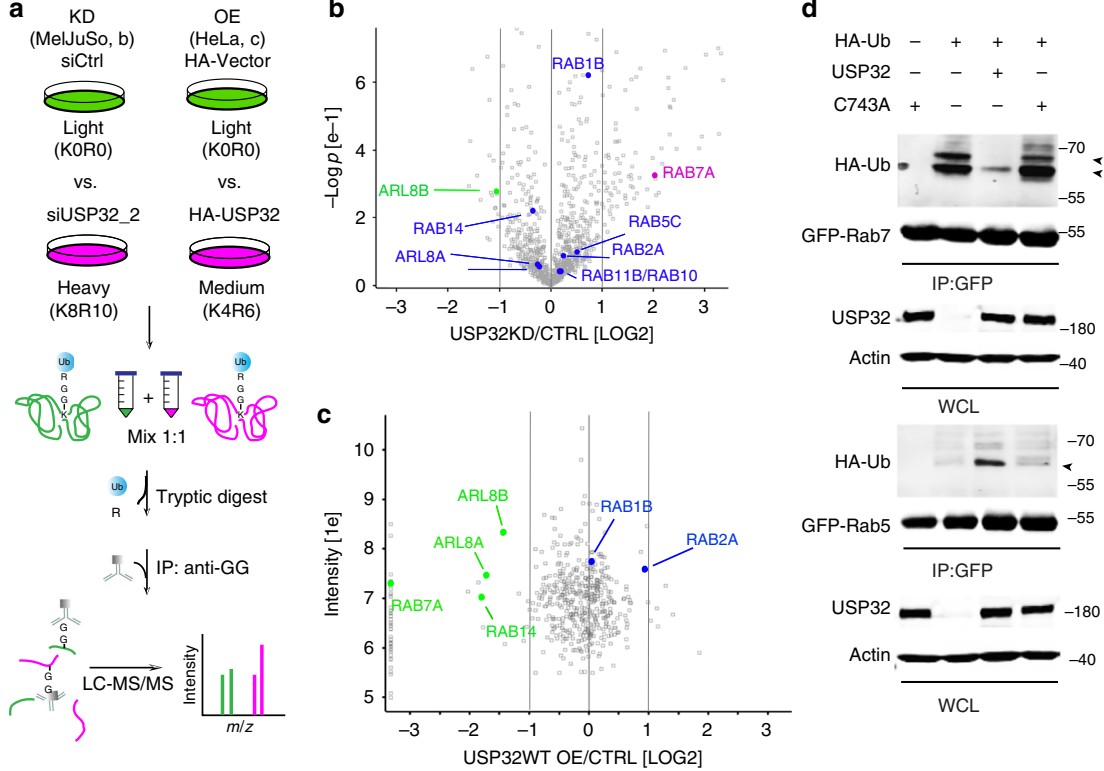

**Fig. 4** Ubiquitome analysis reveals small GTPase Rab7 as a substrate of USP32. **a** Schematic representation of stable isotope labeling of amino acids in cell culture (SILAC)-based quantitative mass spectrometry (LC-MS/MS) workflow used to compare ubiquitylated proteomes of **b** control MelJuso cells (siCtrl, green) vs. those where USP32 was knocked down (KD, siUSP32_2, magenta) and **c** HeLa cells overexpressing (OE) USP32-HA (magenta) vs. vector control (green). Cell growth media types: K0R0, light; K4R6, medium; K8R10, heavy. IP: immunoprecipitation, $m/z$: mass to charge ratio. **b**, **c** Volcano plots comparing abundance of detected peptides carrying a GlyGly (GG) modification expressed as Log 2 ratios of **b** USP32 knockdown (siUSP32_2, KD) vs. control (siCtrl, CTRL), $n = 1$ SILAC sample set independently validated using label-free quantitation (LFQ) with $n = 2$ biologically independent samples or **c** USP32-HA overexpression vs. vector control (Ctrl), $n = 1$ SILAC sample set. Small GTPases implicated in vesicular traffic whose modified peptides were detected are labeled according to their respective Log 2 ratios: magenta >1; green <−1; blue between −1 and 1, not significant. Analysis was performed using MaxQuant and Perseus software tools as described in the Methods under ubiquitome analysis. All mass spectrometry (MS) data can be accessed via the PRIDE repository (PXD011899). **d** Ubiquitylation status of GFP-Rab7 vs. GFP-Rab5 as a function of USP32 catalytic activity. GFP-Rabs, immunoprecipitated (IP) from HEK293T cells coexpressing HA-Ub and either USP32, C743A, or neither, was assessed by immunoblot against HA; WCL: whole cell lysate. See also Supplementary Fig. 5

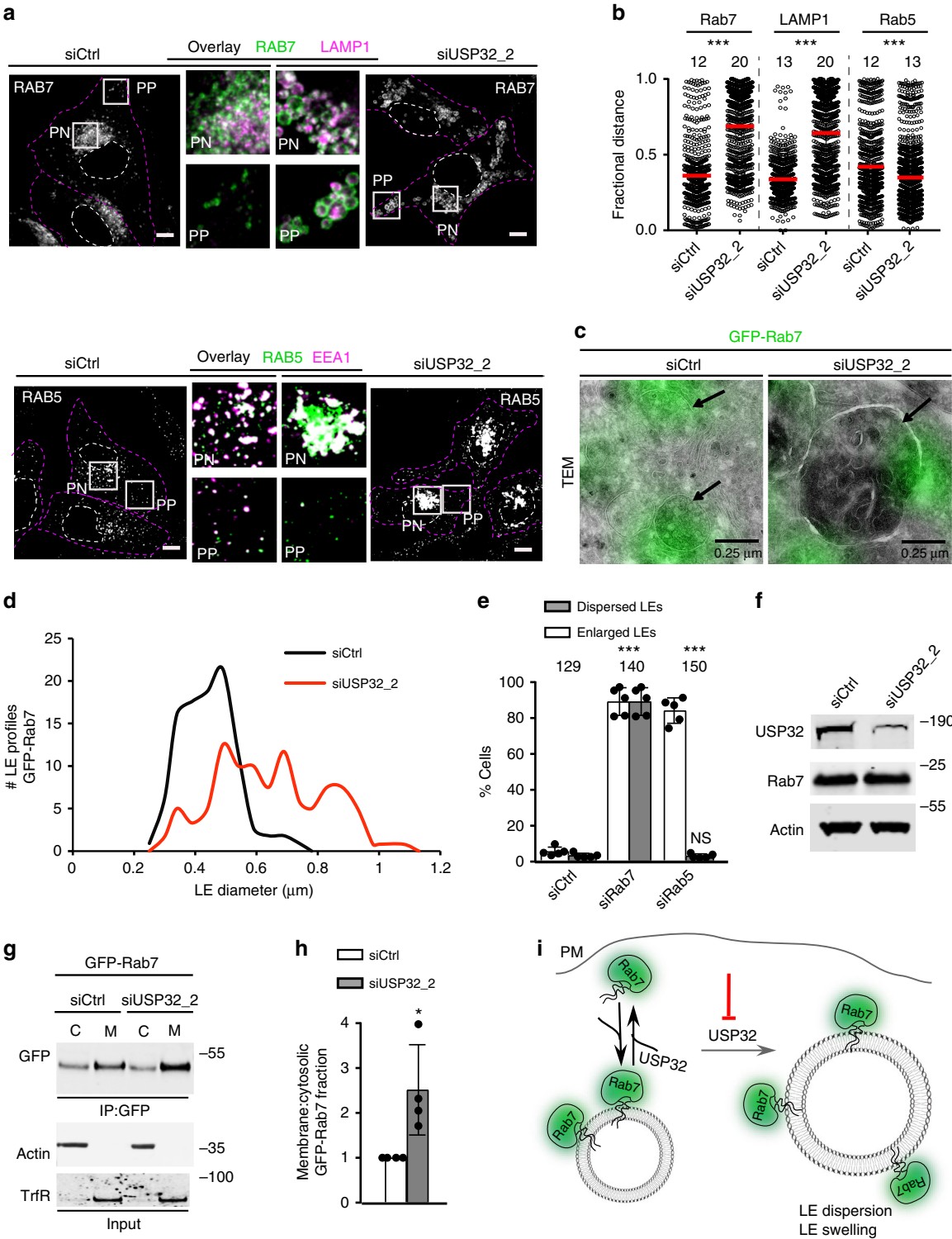

and this alteration was confirmed in a label-free setting (PRIDE dataset identifier PXD011899). Conversely, ubiquitylation on the same residue diminished in the presence of overexpressed USP32 (Fig. 4c). To confirm Rab7 as a bona fide substrate of USP32, we performed an array of validation experiments. A discrete pattern of mono- and/or di-ubiquitin conjugates was observed on a proportion of affinity isolated GFP-Rab7 ectopically expressed in the presence of HA-tagged ubiquitin (HA-Ub). This modification strongly diminished upon coincubation with purified USP32, but not its family member USP30 (Supplementary Fig. 5b–d).

Furthermore, coexpression of GFP-Rab7 in cells with wild-type USP32, but not its enzymatically dead mutant C743A, ablated ubiquitylation on GFP-Rab7 when compared to vector control (Fig. 4d; Supplementary Fig. 5e). In contrast to Rab7 (and in agreement with the proteomic analysis above), no deubiquityla-tion of GFP-Rab5 by USP32 was observed under the same reaction conditions (Fig. 4d; Supplementary Fig. 5e).

**USP32 promotes endosomal transport via the Rab7/RILP axis.** Having established that USP32 can deubiquitylate Rab7 in vitro

**Fig. 5** USP32 regulates the Rab7-positive endosome. **a** Effect of USP32 depletion on Rab7-positive and Rab5-positive endosomes. Top panels: representative confocal images of endogenous Rab7 (white). Boxed region perinuclear (PN) and peripheral (PP) zoom overlays of Rab7 (green) highlight colocalization with late endosome (LE) marker LAMP1 (magenta). Bottom panels: representative confocal images of endogenous Rab5 (white). Boxed PN and PP region overlays of Rab5 (green) highlight colocalization with EE marker EEA1 (magenta). Cell and nuclear boundaries depicted in dashed magenta and white lines, respectively. Scale bars = 10 μm. **b** Rab pixel distribution expressed as fractional distance along a straight line from center of nucleus (0) to the PM (1.0). Red lines: mean, $n = 2$ independent experiments. **c** Alterations in LE morphology in response to USP32 depletion as visualized by correlative light and electron microscopy (CLEM). Overlays of GFP-Rab7 fluorescence (green) and transmission electron micrographs (TEM) are shown, scale bars = 0.25 μm. **d** Comparison of GFP-Rab7-positive LE profiles in control cells (siCtrl, black line) vs. those depleted of USP32 (siUSP32_2, red line), x-axis: LE diameter in μm, y-axis: number of LE profiles. See also Supplementary Fig. 6. **e** LE enlargement (white bars) and/or dispersion (gray bars) in MelJuSo cells depleted of the indicated GTPase (% cells), $n = 4$ or 5 independent experiments as indicated. See also Supplementary Fig. 7a. **f** Effects of USP32 depletion on cellular abundance of endogenous Rab7, as assessed by immunoblot. **g** Analysis of membrane-bound vs. cytosolic fractions of GFP-Rab7 stably expressed in MelJuSo cells with actin and transferrin receptor (TrfR) as loading controls for the cytosolic and membrane fractions, respectively. **h** Ratio of membrane-bound/cytosolic GFP-Rab7 in control cells (siCtrl, white bars) vs. those depleted of USP32 (siUSP32_2, gray bars) normalized to siCtrl, $n = 3$ independent experiments. **i** Schematic illustration of consequences of USP32 depletion on Rab7 membrane-to-cytosol equilibrium. Bar graphs report mean of independent measurements (black circles), error bars reflect ±s.d. Where applicable, total number of cells analyzed per condition appears above each bar/scatter. All significant values were calculated using Student's $t$ test: *$p < 0.05$, ***$p < 0.001$, NS = not significant

and in situ, we sought to understand the interplay between USP32 and Rab7 in endosome biology. Depletion of USP32 incurred dispersion and swelling of vesicular structures decorated with endogenous Rab7 (Fig. 5a, b; Supplementary Fig. 5f), similar to the effects observed for MHC-II and CD63 (Fig. 1b–d). These phenotypes were not seen for endosomes marked by Rab5 (Fig. 5a, b; Supplementary Fig. 5g). Besides the obvious increase in size (Fig. 5c, d; Supplementary Fig. 6a), Rab7-positive LEs affected by loss of USP32 appeared to exhibit abnormal intraluminal content (Supplementary Fig. 6b), consistent with aberrant delivery/retrieval of materials to/from the MVB. Additionally, depletion of Rab7 itself was associated with both mislocalization and enlargement of vesicles carrying MHC-II (Fig. 5e; Supplementary Fig. 7a), implying that Rab7 and USP32 regulate a common biological process. By contrast, loss of Rab5 did not result in swollen MHC-II endosomes (Fig. 5e; Supplementary Fig. 7a), indicating that late compartment swelling is not a general phenotype of endosomal Rab deficiency. Notably, depletion of USP32 did not alter cellular abundance of endogenous Rab7 (Fig. 5f), indicating that the phenotypes caused by USP32 loss are not due to ubiquitylation-dependent degradation of its substrate. On the other hand, silencing USP32 resulted in the accumulation of membrane-bound Rab7, as evidenced by elevated membrane-to-cytosol ratio relative to the control (Fig. 5g, h). This suggested that deubiquitylation of Rab7 by USP32 promotes release of Rab7 from the membrane to the cytosol, where a new functional cycle of this Rab can commence (Fig. 5i).

Based on the observations described above, we hypothesized that diminished availability of cytosolic Rab7 could inhibit Rab7-dependent LE dynamics. We therefore followed the behavior of Rab7-positive endosomes as a function of USP32 in living cells. As expected, under control conditions, vesicles carrying GFP-Rab7 partitioned between a crowded and relatively immobile PN vesicle cloud and a sparsely populated but highly motile peripheral fraction (Fig. 6a–c; Supplementary Movie 9). Loss of USP32 disrupted this PN/PP dichotomy and resulted in an inhibition of LE motility (Fig. 6a–c; Supplementary Movie 10). To test whether these abnormalities arise from insufficient deubiquitylation of Rab7, we examined the ability of its ubiquitylation-deficient mutant to rescue the above phenotype. In addition to mutating USP32 target residue K191 to R, we also mutated neighboring K194 to avoid potential "hopping" of ubiquitin conjugation. The resulting GFP-Rab7-2KR exhibited appreciably less ubiquitylation as compared to its wild-type counterpart and was largely insensitive to coexpression of USP32 (Supplementary Fig. 7b, c). Expression of GFP-Rab7-2KR partially relieved disturbances to the LE compartment organization (dispersion)

and dynamics (motility) sustained under USP32 depletion, as evidenced by the restoration of the PN pool of Rab7-positive LEs and improvement in their motility (Fig. 6a–c; Supplementary Movies 11, 12).

We also noticed that in the absence of USP32 knockdown, expression of GFP-Rab7-2KR exaggerated PN clustering of LEs (Fig. 6a–c), implying that ubiquitylated Rab7 may limit minus-end-direct LE transport. In other words, LE transport toward the nucleus could be preferentially mediated by non-ubiquitylated Rab7. To test this, we examined whether modulating ubiquitylation on Rab7 influences interactions with its effector RILP, known to recruit the dynein motor to LE membranes for transport toward the microtubule minus end (i.e. into the PN region). Indeed, RILP co-isolated better with Rab7-2KR as compared to wild-type Rab7 (Fig. 6d, e). On the other hand, depletion of USP32 slightly diminished complex formation between RILP and wild-type Rab7 (Fig. 6f; Supplementary Fig. 7d). Taken together, these findings demonstrate that RILP favors Rab7 whose C-terminal lysine(s) are not modified with ubiquitin, providing a rationale for how Rab7 mutant lacking these moieties can partly restore LE localization and dynamics in cells compromised for USP32.

**USP32 promotes membrane recycling from the Rab7 endosome.** Among many roles of Rab7 at the LE/MVB[35], facilitated by its canonical effectors such as RILP, this GTPase is also known to partner with the retromer complex to regulate recycling away from late compartments[36–38]. In agreement with a previous report[39], we found that USP32 interacts with VPS35, the principal cargo-selective component of the retromer, and partly localizes to structures positive for both VPS35 and Rab7 (Supplementary Fig. 8a, b). Additionally, silencing either VPS35, or its retromer partner VSP26, recapitulated LE dispersion and swelling observed with disruption of USP32 activity (Fig. 7a, b) similar to the effects observed for silencing of Rab7 (Fig. 5e; Supplementary Fig. 7a). As in the case of Rab7, depletion of USP32 had no effect on protein levels of either VPS35 or VPS26 (Fig. 7c) and, taken together with the observations above, signaled that the retromer complex and USP32 likely function in the same pathway with respect to membrane dynamics at the LE/MVB.

We next examined whether USP32 regulates the interplay between Rab7 and the retromer complex on endosomes. Depletion of USP32 gave rise to swelling of the VPS35-positive vesicular compartment and accumulation of VPS35 on endosomes carrying MHC-II (Fig. 7d, e), suggesting that increased ubiquitylation on Rab7 stabilizes VPS35 on endosomes. This

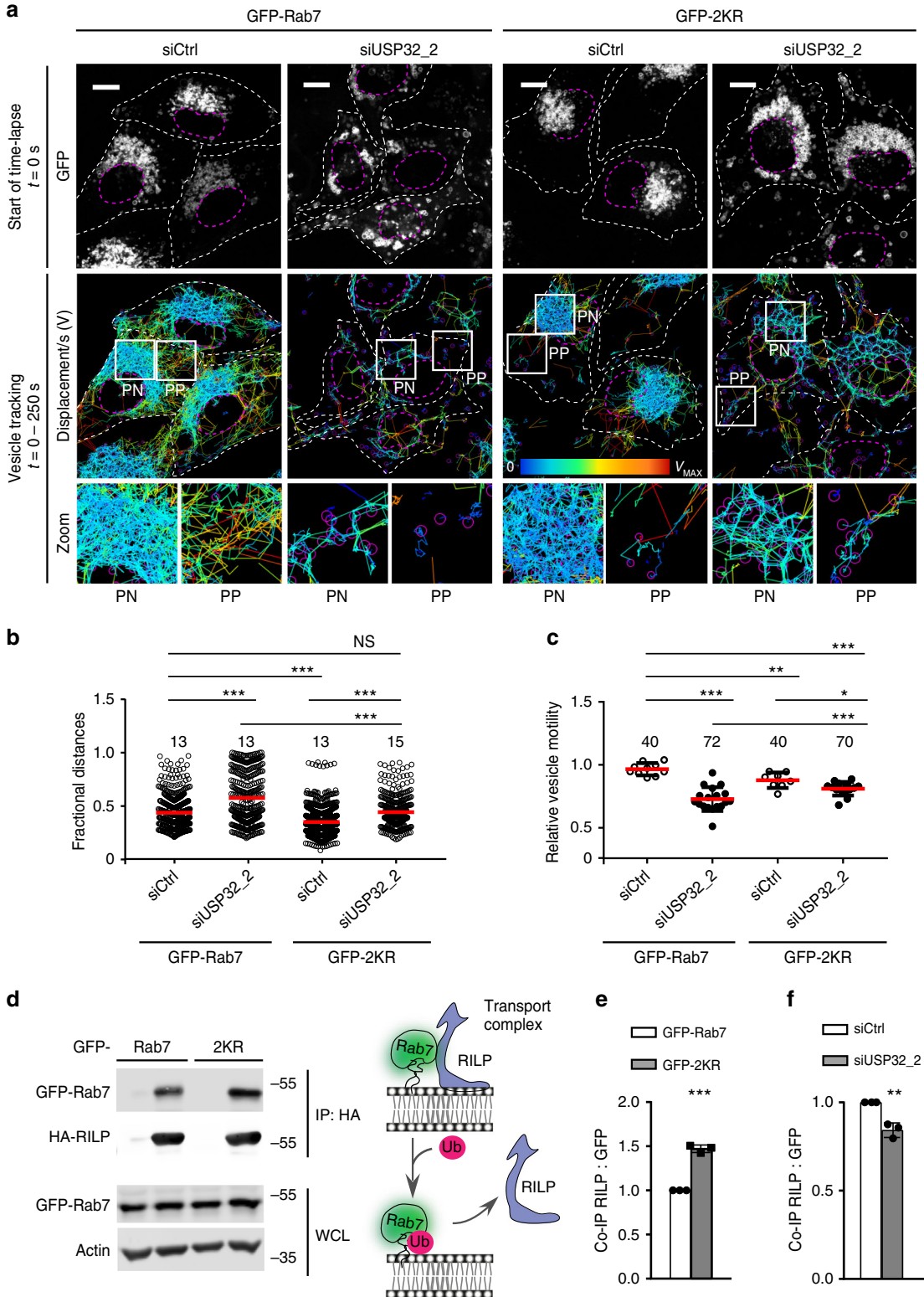

supposition was reinforced by the observation that in cells depleted of endogenous Rab7, endosomes marked by GFP-Rab7-2KR exhibit less contact with structures harboring VPS35, as compared to those with wild-type GFP-Rab7 under the same conditions (Supplementary Fig. 8c–e). To probe whether ubiquitylation of Rab7 affects its relationship with VPS35, we fused a promiscuous biotin ligase domain BirA[40] to the N terminus of Rab7 or its mutant 2KR (following the GFP tag) and

co-expressed these fusion constructs with RFP-VPS35. Following biotin addition, biotinylation of VPS35 was detected in the presence of GFP-BirA-Rab7 above the GFP-BirA control (Fig. 7f), which diminished when GFP-BirA-2KR was expressed instead (Fig. 7f, g). Conversely, depletion of USP32 improved biotinylation of VPS35 in the presence of GFP-BirA-Rab7 (Fig. 7h; Supplementary Fig. 8f). In the same experiment, labeling of endogenous USP32 with biotin was also observed, demonstrating

**Fig. 6** Deubiquitylation of Rab7 by USP32 promotes late endosome transport. **a** Late endosome (LE) organization and dynamics as a function of Rab7 ubiquitylation status. Top panels: representative confocal images of live MelJuSo cells stably expressing GFP-Rab7 or GFP-Rab7-2KR (GFP-2KR) (white) taken at the start of time-lapse, $t = 0$. Scale bars = 10 μm. Bottom panels: vesicle displacement rates depicted on a rainbow color scale (blue: immobile; red: maximum mobility per time interval) tracked over 250 s at 5 s per frame. Cell and nuclear boundaries are depicted in dashed white lines, boxed zoom-ins highlight select perinuclear (PN) and peripheral (PP) regions. **b** Vesicle dispersion expressed as fractional distance of GFP pixels along a straight line from center of nucleus (0) to the PM (1.0). Red lines: mean, $n = 2$ independent experiments. **c** Quantification of vesicle motility calculated using TrackMate for Fiji (for details see the Methods section), $n = 3$ independent experiments. Plots report mean velocities of Lysotracker-positive structures calculated from multicell time-lapses of control cells (siCtrl, open circles) vs. those depleted of USP32 (siUSP32_2, closed circles). See also Supplementary Fig. 7b, c and Movies 9–12. **d–f** Co-immunoprecipitation (Co-IP) of Rab7-interacting protein (RILP) with GFP-Rab7 as a function of Rab7 ubiquitination status. **d** Immunoblots of Co-IP from HEK293T cells transfected and treated as indicated. **e** Quantification of Co-IP for HA-RILP with GFP-Rab7 (white bars) vs. GFP-2KR (gray bars), $n = 3$ independent experiments, is shown along with a schematic summary. **f** Quantification of Co-IP for HA-RILP with GFP-Rab7 from control HEK293T cells (siCtrl, white bars) vs. those depleted of USP32 (siUSP32_2, gray bars) normalized to control, $n = 3$ independent experiments. See also Supplementary Fig. 7d. Bar graphs report mean of independent measurements (black circles), error bars reflect ±s.d. Where applicable, total number of cells analyzed per condition appears above each bar/scatter. All significance was calculated using Student's $t$ test: *$p < 0.05$, **$p < 0.01$, ***$p < 0.001$, NS = not significant

that USP32 is in complex with Rab7 (Supplementary Fig. 8f). Collectively, these findings suggest that ubiquitylated Rab7 attracts VPS35 more so than its unmodified counterpart, and that in the absence of deubiquitylation by USP32, the retromer remains stuck on the Rab7-positive compartment.

If ubiquitylated Rab7 draws in the retromer machinery, we expected that lack of cognate deubiquitylation would then delay fission of tubules recycling from Rab7-positive compartments. Zooming in on the time-lapses of GFP-Rab7/-2KR cells, we noted that in the absence of USP32 buds and tubules emanating from GFP-Rab7-positive LEs frequently failed to separate from their parent endosome—a phenotype rarely observed in control cells (Fig. 8a; Supplementary Movies 13, 14). Notably, neither failure in tubule resolution (Fig. 8b; Supplementary Movies 15, 16), nor morphological aberrations of MVBs incurred in the absence of USP32 (Fig. 8c, d; Supplementary Fig. 9a, b), could be rescued by GFP-Rab7-2KR. Moreover, problems with tubule fission in cell expressing this Rab7 mutant were even observed in the presence of USP32 (Fig. 8b; Supplementary Movies 15, 16), implying that ubiquitylation-deficient Rab7 is unable to support normal membrane retrieval from the MVB. Collectively, these results illustrate that both elevated as well as insufficient levels of Rab7 ubiquitylation disrupt membrane retrieval from the LE. Taken together with the observations on endosomal transport (Fig. 6), these findings support a model wherein deubiquitylation of Rab7 by USP32 exerts multifaceted control over membrane dynamics at the LE/MVB by promoting their intracellular motility as well as enabling efficient recycling from these organelles (Fig. 9).

## Discussion

Ubiquitylation provides key signals in endocytosis by directing cargoes for lysosomal degradation and modulating the functionality of sorting machineries[41]. In turn, deubiquitylation offers a necessary counterforce, imparting spatiotemporal controls to these dynamic processes[42]. Strikingly, out of nearly 100 human DUBs, only a handful have been implicated in the endocytic pathway[43]. To evaluate whether additional DUBs are entrusted with its upkeep, we performed a depletion screen, leading to the identification of USP32 as a potent regulator of the endolysosomal compartment. In cells lacking USP32, these vesicles exhibit an array of architectural and functional defects, including aberrant localization, structure, and motility, as well as compromised resolution of recycling tubules and attenuated cargo proteolysis. Collectively, these deficiencies exemplify the multifaceted impact of USP32 on membrane traffic and underscore the utility of reversible ubiquitylation in the regulation of dynamic cellular processes.

To determine the role of USP32 in endocytosis, a relevant substrate of its DUB activity needed to be identified. Cellular ubiquitome profiling revealed deubiquitylation of Rab7 by USP32 on K191, located in the solvent-exposed flexible C-terminal region of the molecule. Despite the centrality of Rab7 to late endosome biology and prior knowledge of its ubiquitylation on K191[44,45], the consequences of this modification for Rab7 function have not been previously described. We now show that ubiquitylation of Rab7, accumulated under conditions of USP32 depletion, inhibits LE motility and disrupts PN organization of the late compartment. These phenotypes appear to stem from the inhibition of Rab7-mediated minus end-directed transport, as its effector RILP, responsible for recruiting the dynein motor, prefers a Rab7 mutant lacking K191 (along with its neighbor K194). The same mutant restores LE motility and PN localization of Rab7-positive vesicles in the background of USP32 deficiency, lending further support to the notion that (C-terminal) ubiquitylation of Rab7 negatively impacts its function(s) in LE transport. Additionally, while USP32 does not appear to affect the overall abundance of Rab7, its loss leads to the expansion of membrane-associated Rab7 fraction. Together, these observations implicate reversible Rab7 ubiquitylation in the fine-tuning of transport complex assembly and modulation of this GTPase's availability for different functional states, as discussed below.

Besides its key role in transport, Rab7 also curates recycling from the LE in collaboration with the retromer. While Rab7 is known to recruit the retromer complex toward the LE membrane[36,37], how these interactions are regulated to produce a recycling vesicle remains unclear. It was recently suggested that ubiquitylation of Rab7 on K38 positively affects retromer-associated tubulation of LE membranes[14]. Structurally, N and C termini of Rab7 (based on its yeast counterpart Ypt7) are closely juxtaposed in space[46,47], which may allow modifications acquired on K38 and K191 to collaborate. It has also been proposed that coordinated cargo sequestration onto tubules ultimately displaces Rab7 from the membrane, resulting in fission of recycling vesicles from the mother endosome[48]. Given that USP32 associates with the retromer component VPS35[39], it stands to reason that this DUB's activity may impinge at the Rab7/retromer juncture. Indeed, we find that depletion of USP32 traps VPS35 on enlarged LEs and inhibits resolution of buds and tubules emanating from the Rab7-positive endosome, implying that deubiquitylation serves to promote fission of these structures. Furthermore, unlike in the case of LE transport, effects of USP32 loss on the recycling from the Rab7 compartment could not be rescued by the ubiquitylation-compromised Rab7-2KR mutant. Moreover, even in the presence of USP32, cells expressing Rab7-2KR experience a defect in tubule resolution, supporting a model wherein

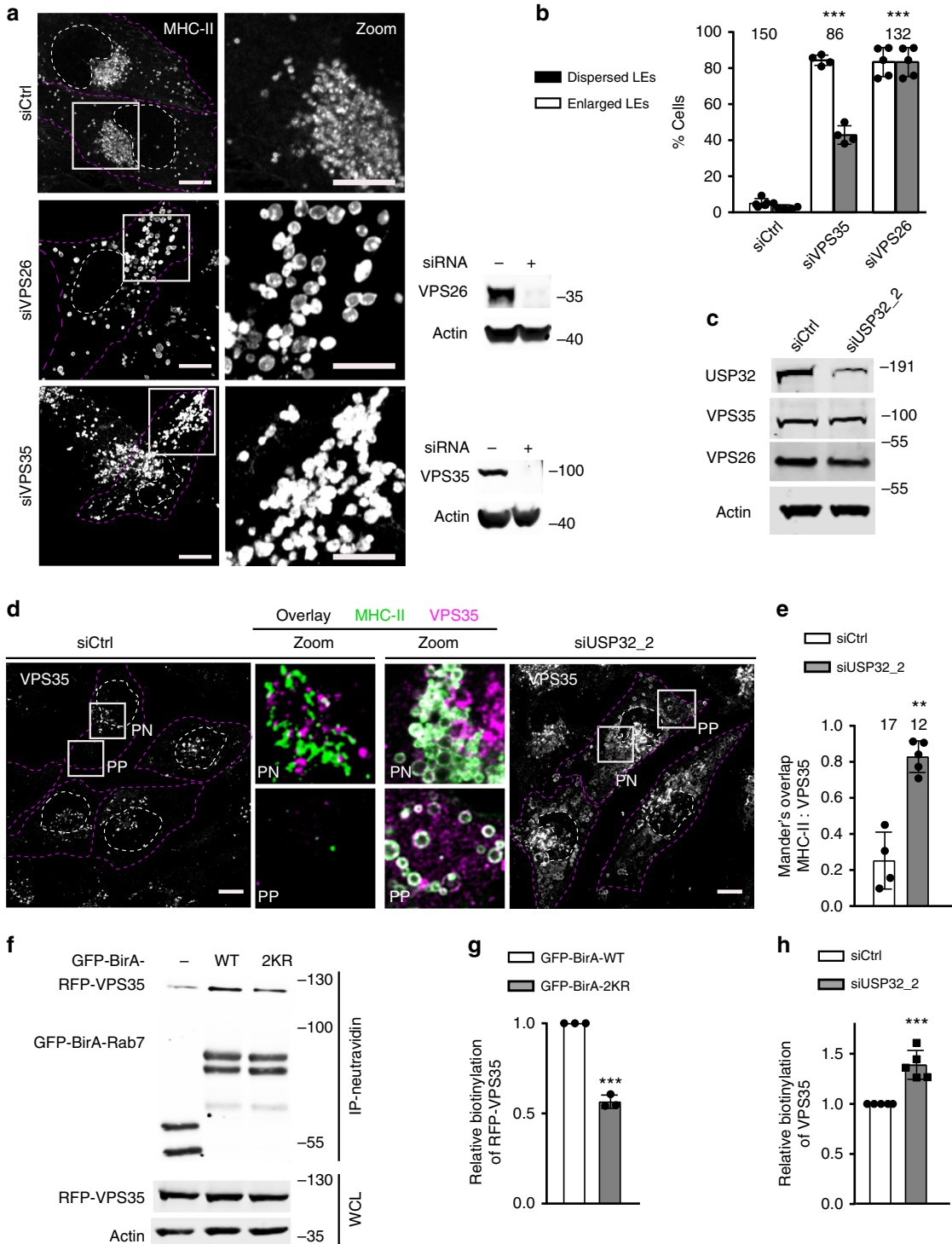

ubiquitylation and subsequent deubiquitylation of Rab7 work in concert to mediate efficient recycling from the LE membrane. Two other DUBs—USP8 and USP7—have also been implicated in various aspects of retromer-associated extraction of cargoes from endosomes[25,49]. Our study reveals an additional regulatory mechanism for this complex and dynamic process, exemplifying the diversity of spatiotemporal regulation through reversible ubiquitylation.

Collectively, our findings establish USP32 as a key component of the molecular repertoire entrusted with guarding the health of the endolysosomal system. Through investigating the effects of

USP32 on its substrate Rab7, we show that Rab7-associated endosomal processes depend not only on its GTP-based state but also on its modification(s) with ubiquitin. In addition to Rab7-driven processes of transport and recycling, the severe nature of USP32 loss-of-function phenotypes leaves open the possibility that additional facets of endosomal traffic and membrane dynamics could be affected. Recently, a number of studies have implicated USP32 in the pathogenesis of various cancers, as well as Parkinson's disease[50–53]. Since the latter, in particular, is associated with defects in endosomal membrane dynamics[54], our study may set the groundwork for understanding and ultimately

**Fig. 7** USP32 promotes retrograde trafficking by way of the retromer complex. **a** Effects of VPS35 and VPS26 depletion on the size and intracellular distribution of late endosomes (LEs). Representative confocal images of fixed MelJuSo cells transfected as indicated and immunostained against major histocompatibility class II (MHC-II) (white) are shown with the corresponding immunoblot analyses; targeting small interfering RNA (siRNA): (+); control siRNA: (−). Cell and nuclear boundaries are depicted in dashed magenta and white lines, respectively. **b** Percent cells harboring dispersed (gray bars) and/ or enlarged (white bars) MHC-II-positive vesicles in response to VPS35 and VPS26 depletion, $n = 3$ independent experiments. **c** Effects of USP32 depletion on cellular abundance of endogenous VPS35 and VPS26, as assessed by immunoblot. **d, e** Effects of Rab7 ubiquitylation status on the retromer compartment. **d** Representative confocal images of fixed MelJuSo cells transferred with the indicated siRNAs and immunostained for endogenous VPS35 (white). Boxed perinuclear (PN) and peripheral (PP) region overlays of VPS35 (magenta) with MHC-II (green) highlight retromer/LE interactions. Cell and nuclear boundaries depicted in dashed magenta and white lines, respectively. **e** Colocalization between VPS35 and MHC-II in control cells (white bars) vs. those depleted of USP32 (gray bars). Plots report Mander's overlap calculated from multicell images (black circles) taken from $n = 3$ independent experiments. See also Supplementary Fig. 8a–e. **f–h** Effect of Rab7 ubiquitylation status on its interaction with VPS35 as measured by proximity-based labeling with biotin. **f** Biotinylation of RFP-VPS35 in the presence of free GFP-BirA (−), GFP-BirA-Rab7 (WT) vs. GFP-BirA-2KR (2KR) assayed in HEK293T cells. **g** Biotinylation of RFP-VPS35 by GFP-BirA-Rab7 (white bar) vs. GFP-BirA-2KR (gray bar) above GFP-BirA background control, $n = 3$ independent experiments. **h** Quantification of endogenous VPS35 and RFP-VPS35 biotinylation (combined) by GFP-BirA-Rab7 above GFP-BirA control in control HeLa cells (siCtrl, white bar) vs. those depleted of USP32 (siUSP32_2, black bar), $n = 5$ independent experiments. Bar graphs report mean, error bars reflect ±s.d. See also Supplementary Fig. 8f. All significant values were calculated using Student's $t$ test: **$p < 0.05$, ***$p < 0.001$, NS = not significant. Sale bars = 10 μm

targeting the involvement of USP32 in this context. On the other hand, association of USP32 with diverse ailments suggests far-reaching cellular activities, possibly independent of Rab7, and elucidating these in the future could provide new avenues for drug discovery.

## Methods

**Cell culture and transfections**. MelJuSo (human melanoma) cells, kindly provided by Prof. G. Riethmuller (LMU, Munich), were cultured in Iscove's modified Dulbecco's medium (IMDM) (Gibco) supplemented with 7.5% fetal calf serum (FCS, Greiner). Human HEK293T (Cat# ATCC® CRL-3216™) and HeLa (Cat# ATCC® CCL-2™) cell lines purchased from ATCC were cultured in Dulbecco's modified Eagle's medium (DMEM) supplemented with 7.5% FCS. MelJuSo cell lines stably expressing GFP-TGN46[24] or GFP-Rab7 were generated by clonal expansion. All the cell lines used in the study were maintained at 37 °C and 5% $CO_2$, routinely scrutinized using morphology analysis and surface marker expression and tested for mycoplasma.

For siRNA transfections, all oligos used in this study were purchased from Dharmacon. Custom siRNA oligos used to target USP32 were as follows: sense-siUSP32#1 CCAGUAAAGGCUACAUCAU and sense-siUSP32#2 GCCUCAGUUACGUGAAUAC[55]. Additionally, the following pre-designed siRNAs were used: siUSP32 pool of 4 (siGENOME Cat# MQ-006080-03-0002), siUSP32_P1 (Cat# D-006080-05-0002), siUSP32_P2 (Cat# D-006080-22-0002), siUSP32_P3 (Cat# D-006080-23-0002), and siUSP32_P4 (Cat# D-006080-24-0002). siRab7 pool of 4 (siGENOME, Cat# MQ-010388-00-0002), siVPS26 (SMARTpool: siGENOME, Cat# M-013195-02-0005), siRab5 (SMARTpool: siGENOME Cat# M-004009-00-0005), and siVPS35 (SMARTpool: siGENOME, Cat# M-010894-00-0005). Rab7-UTR (sense-siRab7-UTR GCUUGGAGAGCU CGGGAGAUU) was also used where indicated. Silencing was performed in MelJuSo and HeLa cells as follows: for 24-well plate format, 50 μL siRNA (500 nM stock) were incubated with 1 μL Dharmafect reagent 1 (Dharmacon) diluted in 49 μL medium without supplements (total volume of 100 μL transfection mix) with gentle shaking for 20 min at room temperature (RT). A total of $15–25 \times 10^3$ cells resuspended in 0.4 mL of growth medium without antibiotics from $37.5 \times 10^3$ cells per mL suspension were added to transfection mixes to a total volume of 0.5 mL per well and cultured for 3 days prior to further analysis. The reaction was scaled using the same component ratios as follows: 35 mm dish—0.4 mL transfection mix plus 1.6 mL cell suspension, 10 cm dish—2.0 mL transfection mix plus 8–10 mL cell suspension.

For DNA transfections, MelJuSo and HeLa cells were seeded to achieve 40–50% confluency the following day. Cells were transfected using Effectene (Qiagen, Cat# 301427) according to the manufacturer's instructions and cultured for 18–24 h prior to further analysis. HEK293T cells were seeded into 60 mm dishes to achieve 50–60% confluence the following day and transfected with PEI (polyethylenimine, Polysciences Inc., Cat# 23966) as follows: 500 μL DMEM medium without supplements was mixed with 36 μL PEI and 12 μg DNA, incubated at RT for 20 min, and added drop-wise to the cells for culturing for 18–24 h prior to further analysis.

**DNA constructs**. USP32-GFP construct[50] was used as a template to amplify USP32 for cloning into mGFP-N1 and 2xHA-N1 mammalian expression vectors using BamHI/SalI restriction sites. For in vitro protein expression, USP32 full-length (FL) and catalytic domain (CD) were cloned into pFastBacNKI-His-3C-LIC-amp using ligation-independent cloning[56]. HA-Ub[57], GFP-Rab7[16], HA-

RILP[58], and TGN46-GFP[24] constructs have been previously described. Catalytically inactive mutant USP32-C743A and C-terminal ubiquitylation-deficient mutant GFP-Rab7-2KR were generated by site-directed mutagenesis (see below). GFP-Rab5 was amplified by PCR using primer containing BamHI/HindIII restriction site and cloned into BglII/HindIII in pEGFP-C1. Plasmid pmr101A-hVPS26 (#17636, Addgene) was used as a template to amplify VPS26 and clone into mRFP-N1 vector using BamHI/SalI restriction sites. VPS35 was amplified by PCR and cloned into mRFP-C1 and mHA-C1 vectors using Asp718/BamHI restriction sites. Rab7 and 2KR mutant were sub-cloned from pEGFP-C1 into pEGFP-BirA*-C1 vector using HindIII restriction. pEGFP-BirA*-C1 was generated by sub-cloning the BirA* from pCDNA3.1 MCS-BirA*1 (#36047, Addgene) using a mega primer approach[59]. GFP-BirA* fragment was generated by two-step PCR and cloned into pEGFP-C1 vector using NheI/BglII restriction sites. For simplicity in labeling, BirA* is referred to simply as BirA throughout the text and figures.

For site-directed mutagenesis, a mixture containing template DNA, 1× Pfu buffer, 10 mM dNTPs, 125 ng forward and reverse primers containing the desired mutation(s), 1 μL Turbo Pfu Polymerase (Agilent), 50 ng template DNA were mixed with autoclaved MilliQ water up to 50 μL reaction volume, and subjected to PCR using the following program: 95 °C for 2 min (95 °C for 50 s; 60 °C for 1 min; 68 °C for 2 min/Kb)×18 cycles; 68 °C for 20 min; 4 °C forever. Forty microliters of amplified product was incubated with 1 μL DpnI (Thermo Fischer Scientific) for 2 h at 37 °C and transformed into competent DH5α. All modified constructs were verified by sequencing. For primer sequences refer to the Supplementary Table 1. All primers were purchased from Sigma-Aldrich.

**Antibodies and fluorescent dyes**. The following antibodies against human antigens were used for confocal microscopy: mouse anti-USP32 A-10 (Santa Cruz Biotechnology, Cat# sc-374465; 1:100), rabbit anti-human HLA-DR (1:300) for MHC-II[21], mouse anti-TrfR (Invitrogen, Cat# 905963A; 1:100), mouse anti-EEA1 (BD Transduction Laboratories, Cat# 610457; 1:100), mouse anti-CD63 NKI-C3 (1:100)[60], rabbit anti-VPS26 (Abcam, Cat# ab181352; 1:100), goat anti-VSP35 (Abcam, Cat# ab10099; 1:50), mouse anti-CI-M6PR (Abcam, Cat# ab2733; 1:100), rabbit anti-Rab7 (Cell Signaling, Cat# 9367s; 1:100), rabbit anti-Rab5 (Cell Signaling, Cat# 3547s; 1:100), mouse anti-LAMP1 (Santa Cruz Biotechnology, Cat# sc-20011; 1:100), rabbit anti-cathepsin D (Abcam, Cat# ab75852, 1:50), and rat anti-HA (Roche, Cat# 3F10, 1:300). For detection by confocal microscopy, primary antibody incubation 1 h at RT was followed by incubation with appropriate secondary goat anti-mouse Alexa 488/568/646 (Thermo Fisher Scientific, Cat# A-11001/A-11004/A-21236, respectively), goat anti-rabbit Alexa 488/568/647 (Thermo Fisher Scientific, Cat# A-11036/A-31571/A-21245, respectively), donkey anti-mouse Alexa 488/647 (Thermo Fisher Scientific, Cat# A-21202/A-31571, respectively), donkey anti-rabbit Alexa 488/647 (Thermo Fisher Scientific, Cat# A-21206/A-31573, respectively), donkey anti-goat Alexa 568 (Thermo Fisher Scientific, Cat# A-11057), and donkey anti-rat Alexa 568 (Biotium, Cat# 20092) in a 1:300 dilution. Lysotracker DeepRed (Thermo Fisher Scientific, Cat# L12492, 1:10,000) was used for detection of acidified compartments by confocal microscopy in live samples. One hundred nanograms per milliliter final concentration of EGF-Alexa 555 (100 μg, Thermo Fisher Scientific, Cat# E35350) was used in endocytosis assays. The following antibodies were used for detection of endogenous and overexpressed proteins by Western blot analysis in a 1:1000 dilution: mouse anti-USP32 (A-10) in (Santa Cruz Biotechnology, Cat# sc-374465), rabbit anti-VPS26 (Abcam, Cat# ab181352), goat anti-VSP35 (Abcam, Cat# ab10099), mouse anti-VPS35 (Santa Cruz Biotechnology, Cat# sc-374372), mouse anti-CI-M6PR (Abcam, Cat# ab2733), rabbit anti-cathepsin D (Abcam, Cat# ab75852), rabbit anti-EGFR (Millipore, Cat # 06-847), mouse anti-phosphotyrosine (4G10 Milli-pore, Cat# 05-321), rabbit anti-mGFP[61], mouse anti-HA (HA.11 (16B12), Covance,

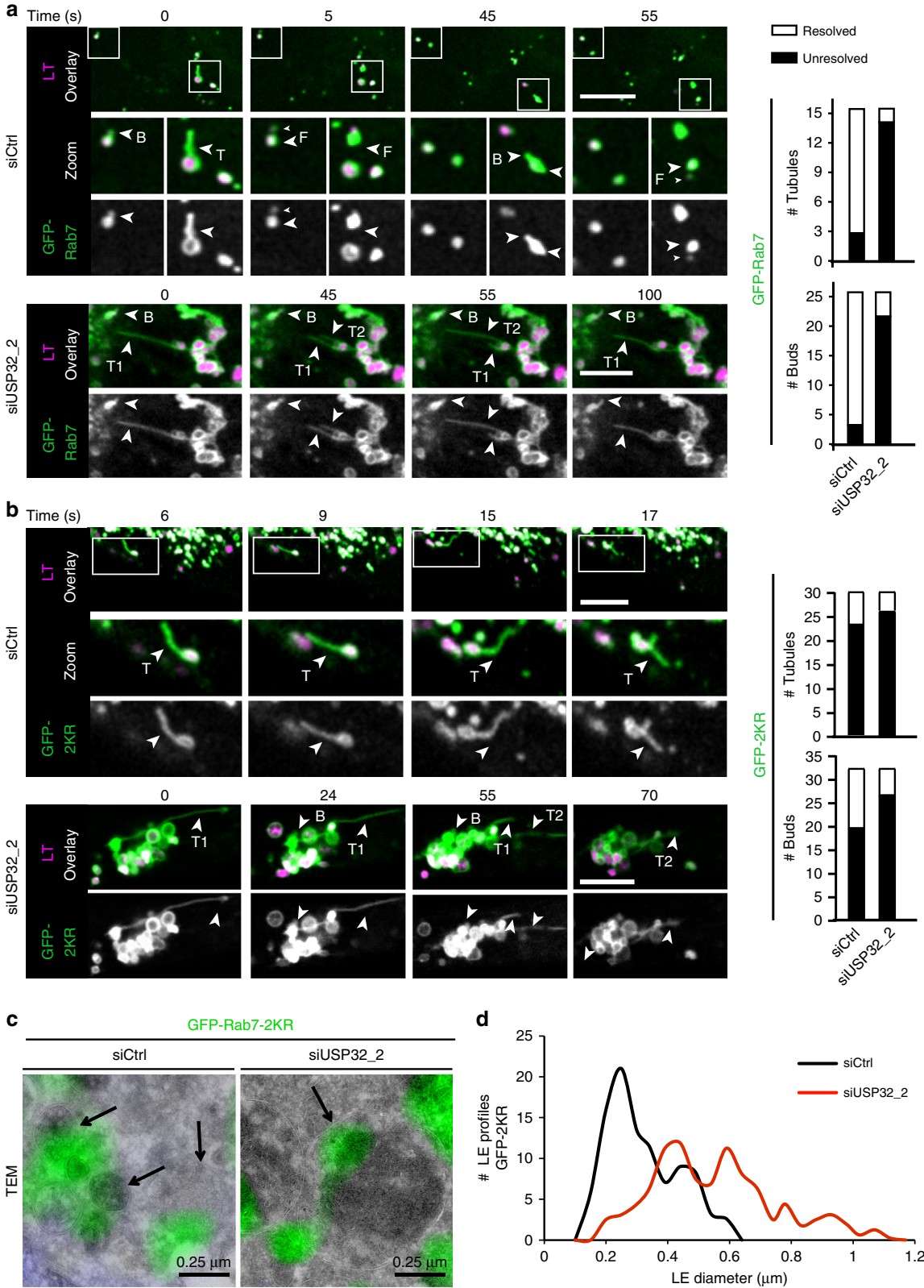

Cat# MMS-101R), and anti-mRFP[61]. Mouse anti-β-actin (Sigma-Aldrich, Cat# A5441) was used as a loading control in a 1:10,000 dilution for Western blot. Secondary IRDye 680LT donkey anti-goat IgG (H + L) (LI-COR, Cat# 926-68024, 1:20,000), IRDye 800CW goat anti-rabbit IgG (H + L) (Li-COR, Cat# 926-32211, 1:5000), IRDye 800CW goat anti-mouse IgG (H + L) (Li-COR, Cat# 926-32210, 1:5000), IRDye 680LT goat anti-rabbit IgG (H + L) (Li-COR, Cat# 926-68021, 1:20,000), and IRDye 680LT goat anti-mouse IgG (H + L) (Li-COR, Cat#

926-68020, 1:20,000) were used for detection using the Odyssey Classic imager (LI-COR).

**siRNA-based DUB screen**. All human DUBs were individually depleted from MelJuSo cells by transfection with pools of siRNAs in 96-well format as follows: 96-well plates—20 μL transfection mix plus 80 μL cell suspension. After 72 h, cells were analyzed for surface expression of peptide-loaded MHC-II by flow cytometry

**Fig. 8** USP32 regulates extraction of membranes from Rab7-positive endosomes. **a**, **b** Bud/tubule resolution from Rab7 endosomes as a function of USP32. Select confocal frame zooms taken from time-lapses of live control (siCtrl) vs. USP32-depleted (siUSP32_2) MelJuSo cells stably expressing (**a**) GFP-Rab7 or (**b**) GFP-2KR (green) labeled with Lysotracker (magenta) are shown. Large arrows point to emerging buds (B) and tubules (T), small arrows point to nascent vesicles formed as a result of fission (F). Quantification: number (#) of resolved (white) and unresolved (black) buds and tubules observed, $n = 2$ independent experiments. Scale bars = 5 μm. See also Supplementary Movies 13–16. **c** Alterations in late endosome (LE) morphology in response to USP32 depletion as visualized by correlative light and electron microscopy (CLEM). GFP-Rab7-2KR (GFP-2KR) fluorescence (green) and transmission electron micrographs (TEMs) are shown; scale bars = 0.25 μm. **d** Comparison of GFP-Rab7-2KR-positive LE profile in siCtrl (black line) and siUSP32_2 (red line); x-axis: LE diameter in μm; y-axis: number of LE profiles. See also Supplementary Fig. 9

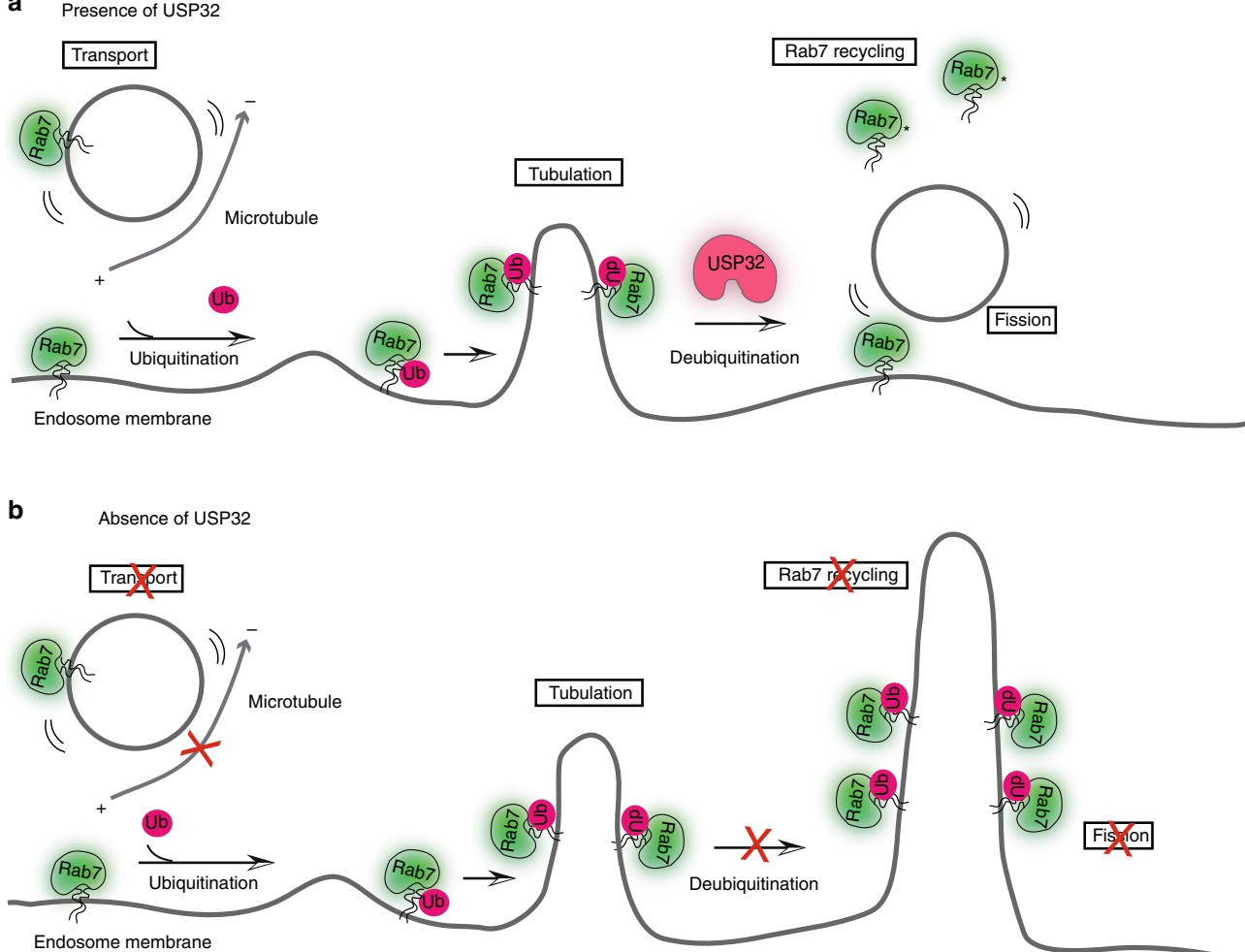

**Fig. 9** Model of USP32 function in Rab7-mediated transport and recycling. **a** In the presence of USP32: non-ubiquitylated Rab7 can efficiently mediate minus end-directed transport, while C-terminal ubiquitination of Rab7 enables the switch to other functions, such as recycling from the late endosome/multi-vesicular body (LE/MVB). In turn, deubiquitylation of Rab7 by USP32 promotes fission of recycling tubules from the mother endosomes and release of Rab7 from the membrane (*refers to GDI (GDP Dissociation Inhibitor) associated with Rab7 upon release). **b** In the absence of USP32: lack of Rab7 deubiquitylation results in failure to resolve recycling tubules and inhibits liberation of Rab7 for subsequent functional cycles. (−): microtubule minus end, leading towards the perinuclear region; (+): microtubule plus end, leading towards the cell periphery

using L243-Cy3 monoclonal antibody. The data were normalized and converted into Z-scores[23]. Those DUBs whose depletion was found to perturb MHC-II surface levels at or above the chosen Z-score threshold of ±3 were subsequently analyzed for intracellular MHC-II phenotypes by microscopy scored on two criteria: (i) size and (ii) distribution of vesicles carrying MHC-II. Those DUB hits, which showed vesicle enlargement and/or redistribution, were followed upon with deconvolution using phenocopy by at least two independent siRNA oligos as minimal criteria.

**Confocal microscopy**. For fluorescence confocal microscopy of fixed samples, cells were seeded into 24-well plates (Costar, Cat# 3524) containing glass coverslips

(Menzel Gläser, Cat# MENZCB00130RAC) and transfected as indicated. Fixation was performed in 3.7% formaldehyde (acid-free, Merck Millipore) in phosphate-buffered saline (PBS) for 20 min. After washing 3× with PBS, samples were permeabilized using 0.1% Triton X-100 (T8787, Sigma-Aldrich) in PBS for 10 min, followed by two washes with PBS. After permeabilization, cells were blocked with 5% (w/v) skim milk powder (LP0031, Oxiod) in PBS for 30 min and incubated with the desired primary antibodies diluted in blocking buffer at dilutions described above for 1 h at RT. Samples were then washed in PBS (three times for 5 min) and incubated with the appropriate secondary anti-rabbit/mouse/rat Alexa-dye-coupled antibodies (Invitrogen) in blocking buffer for 30 min. After washing in PBS (three times for 5 min), cells were mounted using ProLong Gold antifade Mounting medium with DAPI (Life Technologies, Cat# P36941). Samples were imaged using

Leica SP5 or SP8 microscopes equipped with appropriate solid-state lasers, HCX PL 63 times magnification oil emersion objectives and HyD detectors. Data were collected using a digital zoom in the range of 1.5–2.5 in 1024 by 1024 scanning format with line averaging. Post-collection image processing and colocalization analyses were performed using the Fiji software. Colocalization was reported as Mander's overlap or Pierson's coefficient, as indicated. Fractional distances were calculated as follows. Fluorescence intensities (above automated background threshold) were measured along the longest straight line ROI (regions of interest) drawn from the center of a cell's nucleus (fractional distance = 0) to the plasma membrane (fractional distance = 1.0) using the line profile tool in the LAS-AF software, and their absolute distance to the center of the nucleus was expressed relative to the total length of the line[24]. Fractional distances are reported in scatter plots along with the median distance value (red line) within the sample and the total number of cells analyzed.

For fluorescence confocal microscopy of live samples, cells were seeded into 35 mm glass bottom dishes (MatTek, Cat# P35GC-15-14-C) and transfected as indicated. Samples were incubated with Lysotracker DeepRed (1:10,000) 30 min prior to imaging. Imaging was performed on a Leica SP5 microscope with solid-state lasers or Leica SP8 microscope with a white light laser, both equipped with HyD detectors and a climate chamber. Data were collected using ×63 oil immersion objectives in combination with 1.5–2.5 digital zoom in 1024 × 1024 scanning format at 3–5 s intervals with line averaging. Tracking of Lysotracker-positive vesicles was performed using TrackMate for Fiji[24]. Fiji was also used for post-collection image processing.

**Correlative light and electron microscopy.** Cells were fixed at room temperature in 2% paraformaldehyde and 0.2% glutaraldehyde in culture medium for 2 h, embedded in 12% gelatin, infused with 2.3 M sucrose, and frozen in liquid nitrogen. Ultrathin sections (75 nm) of the frozen cells were produced in a Leica UC6 cryo ultra-microtome and transferred onto a 75 mesh titanium EM grid (Agar), carrying a Formvar support film coated with carbon and 100 nm blue fluorescent microspheres (Life Technologies). The sections attached to the grid were rinsed in PBS at 37 °C to remove the gelatin and sucrose, and stained with DAPI (60 ng/mL; Sigma-Aldrich) for 5 min. The grid with the attached sections was imaged in 30% glycerol in a confocal microscope using a ×63 oil immersion objective. After confocal microscopy imaging, the grid with the attached sections was rinsed in distilled water and stained in 1.8% methylcellulose containing 0.4% uranyl acetate and subsequently air-dried before imaging in the TEM. Confocal stacks were deconvolved with theoretical point spread functions using the Huygens Essential deconvolution software (SVI, Hilversum, Netherlands). Electron microscopy images were recorded using a Tecnai 12 TEM (FEI company) equipped with an EAGLE 4 × 4 K digital camera using a magnification of ×13,000. Superimposition and correlation of confocal and electron microscopy images was performed in the Adobe Photoshop software on basis of the signal of the DAPI stain and fluorescent microspheres.

The diameter of GFP-Rab7-positive endosome profiles were measured on 75-nm-thick sections in ImageScope using montages of electron microscopy images produced by stitching software.

**Ubiquitome analysis.** MelJuSo cells were grown in IMDM medium supplemented with 10% FCS, 1% penicilin/streptomycin for label-free quantitation (LFQ) experiments. HeLa or MelJuSo cells were grown either in DMEM or IMDM, respectively, at a minimum of six passages in three batches IMDM media lacking lysine/arginine (Thermo Fisher Scientific) supplemented with 7.5% dialyzed FCS (Thermo Fisher Scientific), 1% penicilin/streptomycin, and either light lysine/arginine (K0R0), medium (4,4,5,5-D4 Lys, $^{13}C_6$Arg—K4R6), or heavy ($^{13}C_6{}^{15}N_2$ Lys, $^{13}C_6{}^{15}N_4$Arg—K8R10) (Thermo Fisher Scientific) amino acids for SILAC-based experiments. USP32 abundance was modulated either by depletion using specific siRNA-mediated knockdown in MelJuSo cells or transient overexpression in HeLa cells. Depletion in MelJuSo cells was performed in two independent ways: (1) using LFQ mode ($n = 2$ biological replicates) and (2) SILAC mode ($n = 1$). Overexpression of USP32-HA vs. vector control was performed in SILAC mode only ($n = 1$). For SILAC experiments, control cells (scrambled siRNA-transfected MelJuSo cells or empty vector transfected HeLa cells) were grown in light media (K0R0); MelJuSo cells transfected with siRNAs targeting USP32 and HeLa cells overexpressing USP32 were grown in heavy media (K8R10 and K4R6, respectively). Approximately $10^8$ cells per condition were used. Cells were lysed in lysis buffer 4 (50 mM Tris-HCl, pH 7.4; 0.5% NP-40; 150 mM NaCl; 20 mM MgCl$_2$) containing 2 mM N-ethylmaleimide. Protein amounts were measured using the BCA Protein assay kit (Pierce, #23225). Samples derived from cells cultured in different isotope-containing media were mixed 1:1 (e.g., 10 mg from control cells and 10 mg from either USP32-depleted or -overexpressing cells), reduced in the presence of dithiothreitol (DTT, 5 mM final concentration), and alkylated using iodoacetamide (20 mM final concentration). Protein precipitation was performed with the methanol/chloroform extraction method. The protein pellet was dissolved in 1 mL of 6 M urea and diluted with 5 mL MilliQ water. Proteins were digested with LysC (Wako, 129-02541, 1:250 (w:w) ~40 mg) overnight at 37 °C following an adjustment of the pH to 7. The second digestion was performed with trypsin (Promega V5113, 1/100 volume of 1 mg/mL trypsin) overnight at 37 °C. Digestion was stopped by the addition of 1% final concentration of trifluoroacetic acid (TFA).

Peptides were purified on a Sep-Pak C18 Column (Waters, WAT020515, 50/pk) and dried using vacuum centrifugation. Peptide enrichment was carried out by di-gly antibody immunoprecipitation according to the manufacturer's instructions (Cell Signalling Technology, Cat# 5562). After immunoprecipitation, enriched peptides were desalted using Sep-Pak C18 columns (Waters, WAT020515) and subsequently dried using vacuum centrifugation. Resulting peptides were resuspended in 10 μL H$_2$O with 2 % acetonitrile and 0.1% formic acid and kept at −20 °C until analysis.

Enriched di-gly peptide samples were analyzed essentially using either a Dionex U3000 system (Thermo Fisher)[62] and an orbitrap Q-Exactive or Fusion Lumos mass spectrometer (Thermo Fisher)[63]. In brief, after peptide loading in 0.1% TFA in 2% acetonitrile onto a trap column (PepMAP C18, 300 μm ×5 mm, 5 μm particle, Thermo), peptides were separated on an easy spray column (PepMAP C18, 75 μm × 500 mm, 2 μm particle, Thermo) with a gradient 2 to 35% acetonitrile in 0.1% formic acid in 5% dimethyl sulfoxide (DMSO). Mass spectrometry (MS) spectra were acquired in profile mode with a resolution of 70,000 with an ion target of $3 \times 10^6$ in the QE mass spectrometer. The Universal Method was used for the Fusion Lumos mass spectrometer (Thermo). The QE instrument was set to pick the 15 most intense features for subsequent tandem mass spectrometry analysis at a resolution of 17,500, a maximum acquisition time of 128 ms, an AGC target of $1 \times 10^5$, an isolation width of 1.6 Th, and a dynamic exclusion of 27 s (Fusion Lumos 12 s). The orbitrap Fusion Lumos was used for SILAC samples and QE for LFQ samples. Raw data were converted to Mascot generic files using msconvert[63], and database searches were performed with MASCOT[64]. Alternatively, for quantitative analysis of all SILAC samples combined, raw MS data were processed using MaxQuant (v1.5.3.1) and subsequently analyzed using the Perseus software (v1.5.3.1). The MS data have been deposited to the ProteomeXchange Consortium via the PRIDE partner repository with the dataset identifier PXD011899.

**Ubiquitination assays.** Ubiquitination status of GFP-tagged proteins was assessed by using ubiquitination assay[24]. HEK293T cells, transfected with HA-ubiquitin, GFP-substrates, and USP32 or its mutant as indicated, were lysed in 300 μL lysis buffer 1 (50 mM Tris-HCl, pH 7.5, 150 mM NaCl, 5 mM ethylenediaminete-traacetic acid (EDTA), 0.5% Triton X-100, 10 mM N-methyl maleimide (general DUB inhibitor diluted in DMSO, freshly added) and protease inhibitors (Roche Diagnostics, EDTA-free, freshly added) by scraping. Then, 100 μL lysis buffer 2 (100 mM Tris-HCl, pH 8.0, 1 mM EDTA, and 2% SDS) were added to the crude lysates; samples were sonicated (Fisher Scientific FB120 Sonic Dismembrator, 3 pulses, amplitude 40%) and SDS was subsequently diluted by bringing sample volume to 1 mL with lysis buffer 1. After centrifugation (20 min, 4 °C, 20,817× $g$), lysates were incubated with 6 μL GFP_Trap_A beads (Chromotek) overnight at 4 °C. Beads were washed three times with lysis buffer. During the fourth washing step, 30 μL Protein G4 fast flow (GE Healthcare) was added and all liquid was removed prior to the addition of SDS sample buffer (containing 10 mM DTT). Proteins were denatured by heating at 95 °C for 15 min, subjected to 8% SDS-PAGE, and detected by Western blotting, as indicated.

**Protein expression and purification.** *Spodoptera frugiperda* Sf9 insect cells were used as hosts for the baculovirus. SF900 II SFM medium (Gibco, Cat# 10902096) containing 1% penicillin–streptomycin (Gibco, Cat# 15140122) was used for insect cell culturing. Suspension insect cells were grown under serum-free conditions at 27 °C with shaking. Sf9 insect cells were transfected with 10 μg bacmid DNA purified from DH10Bac-competent cells (transformed with pFastNKI-his3C-LIC-USP32-FL and pFastNKI-his3C-LIC-USP32-CD plasmids) using Cellfectin II reagent (Gibco, Cat# 10362100). Three days after transfection, P1 baculovirus stock was collected from the culture medium. A total of $1 \times 10^6$ cells/mL in 30 mL medium was infected with P1 baculovirus stock to prepare P2 baculovirus stock. Baculovirus stocks were stored at 4 °C and protected from light. A total of $1 \times 10^6$ Sf9 insect cells were infected using a low MOI (multiplicity of infection ratio of infectious virus particles) to infect the cells. The cells were harvested 72 h after the baculovirus infection. Cells were lysed with lysis buffer 3 (20 mM Tris, pH 8.0, 500 mM NaCl, 5 mM β-mercaptoethanol, 10 mM imidazole, and protease inhibitor cocktail) and sonication. The lysates were centrifuged at 21,000 × $g$ for 30 min at 4 °C. The supernatants were incubated with washed Talon metal affinity resin (Clontech Inc., Palo Alto, CA, USA) for 20 min at 4 °C and the beads were then washed with wash buffer (20 mM Tris, pH 8.0, 500 mM NaCl, and 5 mM β-mercaptoethanol, and 10 mM imidazole). Protein was eluted with elution buffer containing 20 mM Tris, pH 8.0, 500 mM NaCl, 5 mM β-mercaptoethanol, and 250 mM imidazole. Protein was dialyzed to remove imidazol and purified further with a size exclusion column (S200 16/60 column) using a AktaPrime purifier. All proteins were stored at −80 °C.

**Enzyme activity assays.** Di-ubiquitin hydrolysis assay was performed in a buffer containing 50 mM HEPES (pH 7.5), 100 mM NaCl, 5 mM DTT, and 75 nM enzyme: USP32-FL or USP32-CD and 5 μg synthetic di-ubiquitin of specific linkage generated in our lab (M1, K6, K11, K27, K29, K33, K48, and K63) for each reaction[64]. Hydrolysis reactions were kept at 37 °C for 0, 30, and 60 min and stopped by the addition of SDS-containing sample loading buffer and heat denaturation. Samples were analyzed by SDS-PAGE on 4–12 % Bis-Tris NuPage gel

(Invitrogen). Gels were stained with Coomassie blue and detection was performed using molecular imager ChemiDoc XRS + system with the Image lab software (Bio-Rad).

Ubiquitin-fluorescence polarization (Ub-FP) assay was performed with a range of concentrations of USP32-FL and USP32-CD to determine a suitable enzyme concentration[32]. Diluted enzymes were prepared with reaction buffer 1 (20 mM Tris-HCl, pH 7.5, 100 mM NaCl, 5 mM DTT, 0.5 mg/mL bovine γ-globulin, and 10 mg/mL CHAPS (3-[(3-cholamidopropyl)dimethylammonio]-1-propanesulfonate) and 400 nM final concentration of substrate (TAMRA K (Ub) G) prepared with the same buffer. After preparation of enzyme and buffer, the reaction was started by the addition of substrate. The Ub-FP assay was performed on a Perkin-Elmer Wallac EnVisin2010 multilabel reader equipped with 531 nm excitation and 579 nm emission filters. Non-binding surface flat-bottomed low flange black 384-well plates were used for the assay. Data were analyzed using the Microsoft Excel and GraphPad Prism5 software.

Activity-based probe assays were performed as follows. Cy5-Ub-Prg probe[65] was added a final concentration of 0.5 mg/mL to either 100 nM of purified USP32 or USP30, or incubated with clarified lysates of HEK293T cells transfected with HA-N1 vector, USP32-HA or C743A-HA. Reactions were incubated at 37 °C for 30 min and stopped by the addition reducing sample loading buffer (Invitrogen, Cat# N0007). Samples were resolved on 4–12 % Bis-Tris NuPage gel (Invitrogen, Cat# NP0323BOX) in MOPS buffer (Invitrogen, Cat# NP0001) and visualized on a Typhoon 9500 Scanner Image System using 635 nM excitation and 685 nM emission filters or on a Perkin-Elmer ProExpress 2D Proteomic Image System.

**Subcellular fractionation**. MelJuSo cells stably expressing GFP-Rab7 were transfected in 10 cm dishes with the indicated siRNA. Seventy-two hours after transfection, cells were harvested in lysis buffer containing 20 mM Tris-HCl (pH 7.5), 1 mM MgCl$_2$, and protease inhibitors (Roche, complete EDTA-free, Cat# 05056489001), and cell suspension was passed through a 21 G needle 10 times or more (until near complete lysis) using a 5 mL syringe. Samples were centrifuged at 1000×g for 10 min. The supernatant was transferred into fresh tubes and centrifuged at 2000 × g for 20 min. The resulting supernatant (containing membrane and cytosolic fraction) was transferred into fresh tubes and centrifuged at 20,000 × g for 30 min. The supernatant containing the cytosolic fraction was transferred into a new tube, and the pellet was resuspended with a lysis buffer containing 0.5% NP-40, 20 mM Tris-HCl (pH 7.5), 1 mM MgCl$_2$, and protease inhibitors, and then incubated on ice for 20 min. Resulting cytosolic and membrane fractions were incubated with 6 μL GFP_Trap_A beads (Chromotek) overnight at 4 °C. Beads were washed three times with lysis buffer. During the fourth washing step, 30 μL Protein G4 fast flow beads (GE Healthcare) was added and all liquid was removed prior to the addition of SDS sample buffer containing 10 mM DTT. Proteins were denatured by heating at 95 °C for 15 min, subjected to 8% SDS-PAGE, and detected by Western blotting, as indicated.

**Co-immunoprecipitation**. HEK293T cells were lysed for 20 min in a lysis buffer containing 0.8% NP-40, 50 mM NaCl, 50 mM Tris-HCl (pH 8.0), 5 mM MgCl$_2$, and protease inhibitors (Roche, complete EDTA-free, Cat# 05056489001). The supernatant after spinning (20 min, 4 °C, 20,817 × g) was incubated with respective antibodies by rotation at 4 °C for 1 h. Protein G4 fast flow (GE Healthcare) beads were then added to the supernatant and incubated by rotating at 4 °C for 4 h. Beads were washed four times in wash buffer containing 0.08% NP-40, 150 mM NaCl, 50 mM Tris-HCl (pH 8.0), and 5 mM MgCl$_2$. After completely removing the washing buffer, SDS sample buffer (containing 10 mM DTT) was added to the beads followed by 15 min incubation at 95 °C. Co-immunoprecipitated proteins were separated by SDS-PAGE for Western blotting and detection by antibody staining. The obtained signals were detected by Odyssey imager.

**Proximity-based labeling**. HEK293T or HeLa cells were transfected with either GFP-BirA vector control or GFP-BirA-Rab7/2KR either directly or after 48 h of siRNA transfection. After 24 h transfection, cells were treated with 50 μM final concentration of biotin (Sigma, Cat# B4639) for 3 h. Cells were lysed for 20 min in lysis buffer containing 0.8% NP-40, 50 mM NaCl, 50 mM Tris-HCl (pH 8.0), 5 mM MgCl$_2$, and protease inhibitors (Roche, complete EDTA-free, Cat# 05056489001). SDS (0.5%) was added to the supernatant after spinning (20 min, 4 °C, 20,817 × g). High-capacity neutravidin beads (Thermo Scientific, Cat# 29202) were then added to the beads and incubated by rotating at 4 °C for overnight. Beads were washed four times in wash buffer containing 0.8% NP-40, 150 mM NaCl, 50 mM Tris-HCl (pH 8.0), 0.5% SDS, and 5 mM MgCl$_2$. After completely removing the washing buffer, SDS sample buffer (containing 10 mM DTT) was added to the beads, followed by 15 min incubation at 95 °C. Co-immunoprecipitated proteins were separated by SDS-PAGE for Western blotting and detection by antibody staining. The obtained signals were detected by Odyssey imager.

**EGFR degradation**. Ligand-induced turnover of EGFR was performed using 25 ng/mL EGF[57]. HeLa cells were transfected with siCtrl and siUSP32 oligos as indicated. After 72 h incubation, cells were serum starved with serum-free media for 3 h and incubated with 25 ng/mL EGF for 0, 30, 60, or 120 min. Receptor abundance at each indicated time-point following stimulation was quantified

relative to actin and expressed as a fraction of EGFR at $t = 0$ for each condition. Receptor phosphorylation was expressed relative to the maximal activation detected in control cells (pY at $t = 30$).

**M6PR internalization and recycling**. After 72 h of siRNA transfection, HeLa cells were incubated at 37 °C with serum-free DMEM containing 10 μg/mL mouse-cation-independent M6PR antibody for 1 h. Cells were then quickly rinsed with PBS and stained with internalized antibody using anti-mouse IgG. The percentage of cells with dispersed M6PR was quantified by selecting different areas from three independent experiments.

**SDS-PAGE and immunoblotting**. Samples were separated by an 8% SDS-PAGE. Proteins were transferred to a nitrocellulose membrane (Protan BA85, 0.45 μm, GE Healthcare) at 300 mA for 2.5 h. The membranes were blocked in 5% milk (skim milk powder, LP0031, Oxiod) in 1× PBS (P1379, Sigma-Aldrich), incubated with a primary antibody diluted in 5% milk in 0.1% PBS-Tween 20 (PBST) for 1 h, washed three times for 10 min in 0.1% PBST, incubated with the secondary antibody diluted in 5% milk in 0.1% PBST for 30 min, and washed three times again in 0.1% PBST. The signal was detected using direct imaging by the Odyssey Classic imager (LI-COR). Intensity of bands was quantified using the Image Studio software. Unmodified blots corresponding to key experiments presented in Figs. 1–8 can be found in Supplementary Figs. 10 and 11.

**Statistical analysis**. Statistical evaluations report on Student's $t$ test (two-tailed distribution) with *$p < 0.05$, **$p < 0.01$, and ***$p < 0.001$, NS: not significant). All error bars correspond to the mean ± SD.

**Reporting summary**. Further information on experimental design is available in the Nature Research Reporting Summary linked to this article.

## Data availability

All relevant data are available from the authors. The mass spectrometry data associated with Fig. 4a–c have been deposited to the ProteomeXchange Consortium via the PRIDE partner repository under the accession code PXD011899.

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

## Acknowledgements

We are grateful to P. Celie and M. Stadnik from the NKI protein facility for USP32 expression and purification from insect cells. We thank B. Morris for providing siRNA constructs and D.S. Hameed for synthesis of the di-ubiquitin chains. We also thank L. Oomen and L. Brocks of the NKI and A.M.A. van der Laan and L. Voortman of LUMC for microscopy facility support. We are grateful to D. Flierman for critical discussions. This work was supported by the UPStream network funded by a Marie Curie Initial Training Network (ITN) grant from the European Union (FP7A-PEOPLE-2011-ITN) and by an Innovational Research Incentives Scheme Vici grant from the Netherlands Foundation for Scientific Research (N.W.O.) to H.O. (724.013.002), as well as by ERC Advanced grants to J.N. A.S. was supported by the European Cooperation in Science and Technology (COST) with a short-term scientific mission grant (STSM) to perform ubiquitome experiments.

## Author contributions

A.S. and I.B. designed the study and performed and analyzed most of the experiments. R.H.W. performed siRNA-based screen for DUBs. E.B. and R.I.K. carried out correlative light and electron microscopy studies, and H.J. performed electron microscopy on

prepared samples. B.M.K. and R.K. carried out quantitative mass spectrometry experiments and analyzed the ubiquitome data using prepared samples. A.E.E.-B. coordinated biochemical experiments. J.J.A prepared GFP-BirA-Rab7 constructs for biotinylation assay. J.N. advised on the project. H.O. coordinated the study. I.B. and A.S. wrote the manuscript with the help of J.N. and H.O.

## Additional information

**Competing interests:** The authors declare no competing interests.

