## [Peer Review File · Nature Communications]

Reviewers' comments:

Reviewer #1 (Remarks to the Author):

The manuscript of Sapmaz et al. describes the identification of USP32 as an endosome-specific deubiquitination enzyme. Based on altered endosome morphology, the authors provide evidence that USP32 is involved in deubiquitination of Rab7, and find that loss of USP32 causes dispersion and swelling of Rab7-positive vesicles and reduced retromer-mediated recycling. The compensation of this phenotype requires active USP32.

The authors provide a comprehensive and clear study that reveals a critical role of USP32 in endosomal biogenesis. Their data is supported by multiple colocalization and rescue experiments with USP32. Despite my enthusiasm for these findings, the study has one major deficit as it does not unravel the direct role of Rab7 ubiquitination and deubiquitination, but deduces this indirectly. All conclusions are based on the consequences of USP32 depletion rather than Rab7 mutant analyses. This keeps the nagging doubt that another target of USP32 could be responsible for the main function.

Specific issues:

- 1) The authors state in their title that USP32 acts through deubiquitination of Rab7. However, they only show this indirectly. It is not clear if USP32 acts only on Rab7 or has also other targets. I find it very critical that the authors analyze a Rab7 K1912R mutant in their study to test if retromer interaction and endosomal recycling is restored. This would also test if Rab7 is potentially hyperactive.
- 2) How do cells react to USP32 overexpression? If I interpret the data right, then this has only been analyzed in Figure 3, but here also only with respect to ubiquitination (or better the loss of ubiquitination). I would expect that overexpression would mimick the Rab7 KR mutant.
- 3) The authors observe that endosomes are enlarged (Figure 4a) in the absence of USP32. Is the localization of Rab5 impaired under these conditions? If Rab7 is localized to endosomes, it is still in the GTP-form and cannot bind retromer? This would suggest that only one of its functions is impaired (recycling), yet Rab7 may still contribute to fusion.

Reviewer #2 (Remarks to the Author):

Sapmaz et al., "USP32 mediates retrograde trafficking through deubiquitination of Rab7".

This is an interesting study with lots of nice data. It does however need revision before publication and set out below are the changes I think it needs. I have only one suggestion for additional experimental data, other changes are restricted to text and presentation of the figures.

1. The authors should determine if USP32 deubiquitination of Rab7 regulates the association of the retromer cargo-selective complex with Rab7. This can be achieved by expressing GFP-Rab7 and then immunoprecipitating GFP-Rab7 with its interacting proteins in cells treated with siRNA to knockdown USP32, or where USP32 is over expressed. Addition of this data will significantly strengthen the manuscript in my view.
2. I think the title should be, "USP32 regulates retrograde trafficking through deubiquitination of Rab7" as this better describes the role of USP32.
3. Some of the figures are poor. The micrographs are often overly busy with small images covered in lines etc. The position of the inset boxes sometimes does not correspond to the enlarged image. For example, the M6PR zooms in Fig 1 are showing a slightly different region on the original micrograph than indicated by the inset box. All of the figures with inset boxes need to be carefully

checked to make sure that the zoomed image is correct. The presence of two M6PR zoom boxes in the diagram that is part of Fig 1 is confusing as it makes five zoom boxes but only four images are part of figure 1a. Why does the lysosome in Figure 1 have a double membrane? Is it pretending to be an autophagosome? Overall, I was somewhat disappointed with the presentation of the data, it felt at times like a triumph of style over substance which is a shame as the data is generally good.

4. In addition to ref 24, the authors should also cite Rojas et al., (2008) (PMID 18981234) as evidence for Rab7 associating with and regulating the retromer complex.
5. Line 228, "indespensible" is misspelled.
6. Line 247, I am at a loss to understand why the authors have cited refs 48,49 in a sentence describing the M6PR recycling back to the TGN via the retromer complex.
7. In figure 5e, why is the MHC II used as a cargo for retrograde trafficking? Why not investigate the M6PR recycling?
8. The inset boxes in Figure 6a don't correspond to the zooms very well.

Reviewer #3 (Remarks to the Author):

The paper describes follow-up studies from an siRNA screen which identified the deubiquitinase (DUB) as a regulator of late endosome distribution in MeIJuso studies.

GFP-USP32 is mainly associated with the secretory pathway and concentrated in the TGN, yet a proteomic screen for potential USP32 substrates identified the late endosomal GTPase rab7 to be a potential substrate. USP32 depletion is also argued to disrupt recycling of ciM6PR from endosomes to Golgi, a process which is known to be rab7-dependent.

The overall results are certainly plausible but not entirely convincing at this stage. Ultimately there is no mechanism elucidated but a suggestion it may be through rab7 that is poorly supported. The paper is idiosyncratically written and the description of the state of the art vis a vis the endosomal system leaves much to be desired. Methods are lacking essential detail. There are also some experimental areas where quality could be improved. In my opinion not of sufficient general interest for Nature Comms.

Ubiquitination of rab7 itself is already known from the cited paper (ref 31) and a further 45 papers identify the same Lys191 site according to phosphosite. No attempt is made to probe the function of this modification and it is not even clear if this is enriched on the membrane associated fraction. Nor do we ever see any endogenous rab7 blots. Interaction with Vps35 has previously been shown by the Harper lab.

Two other DUBs have been previously linked to M6PR recycling. Hao et al. claim USP7 is required for regulation of the WASH complex (Mol. Cell 2015), whilst Macdonald et al. show that depletion of USP8 also blocks M6PR recycling and that this may be linked to the stability of ESCRT-0. I was surprised that neither of these studies is referred to. Furthermore the authors need to show awareness of recent papers suggesting that ciM6PR retrieval is retromer independent.

Specific Comments:

Details/data of the siRNA screen are lacking. Was the original screen performed with pooled oligos? how many hits? Do all oligos used in the original screen deconvolute.

Fig. 1- how is dispersion quantitated? error bars and statistical tests are not defined throughout. Is this the best way to quantitate in part C- what about some visual representation of pixels.

Fig 2. I could not find details of the anti-USP32 antibody. What is it? How was it validated for IF and em. In other words need to show siRNA controls.

d. the re-expressed USP32 does not look like the same distribution at the endogenous-is this correct. I could not find details of the siRNA resistant construct.

Figure 3. Schematic needs to highlight these experiments use different cell types. Rab7A is not the most sensitive protein to USP32 depletion- what are these other proteins? There should be a spreadsheet for these experiments so readers can look up individual sites. Raw data should be made available through repositories. To my mind this experiment should be done with both oligos. d. need to be more careful and consistent with labelling they are looking at GFP-RAB7A in all cases. Why not look at endogenous rab7.

Figure 5. Endogenous staining for USP32 in HeLa cells does not look like the TGN- how does it compare with epitope tagged?
panel e looks over-exposed. Error bars and stats?

Figure 6 legend EGFR phosphosite needs to be specified.
panel g/h. The data showing a major block to EGFR degradation contradict the findings of Savio et al. Curr Biol (2015)- need to comment on this and also show with a second oligo.
They should ideally measure cathepsin D secretion into the medium.

Reviewer #1 (Remarks to the Author):

The manuscript of Sapmaz et al. describes the identification of USP32 as an endosome-specific deubiquitination enzyme. Based on altered endosome morphology, the authors provide evidence that USP32 is involved in deubiquitination of Rab7, and find that loss of USP32 causes dispersion and swelling of Rab7-positive vesicles and reduced retromer-mediated recycling. The compensation of this phenotype requires active USP32.

The authors provide a comprehensive and clear study that reveals a critical role of USP32 in endosomal biogenesis. Their data is supported by multiple colocalization and rescue experiments with USP32. Despite my enthusiasm for these findings, the study has one major deficit as it does not unravel the direct role of Rab7 ubiquitination and deubiquitination, but deduces this indirectly. All conclusions are based on the consequences of USP32 depletion rather than Rab7 mutant analyses. This keeps the nagging doubt that another target of USP32 could be responsible for the main function.

We thank the Reviewer for the complimentary assessment of our study and the data presented therein. We have performed the suggested analyses and fully agree that probing the consequences of reversible ubiquitylation of Rab7 with respect to its functions in endosome biology greatly strengthens the manuscript. The new data can be found in Fig. 4, which deals with the effects of Rab7 ubiquitylation and deubiquitylation on Late endosome (LE) transport, and Fig. 5 (extended from former Fig. 4) addressing Rab7-mediated recycling from the LE. Based on this expanded characterization of the role(s) of reversible Rab7 modification with ubiquitin, a model summarizing the findings of the manuscript is proposed in Fig. 8. The details of the new findings are discussed below. We extend our appreciation to the Reviewer for valuable suggestions to improve this work.

Specific issues:

1) The authors state in their title that USP32 acts through deubiquitination of Rab7. However, they only show this indirectly. It is not clear if USP32 acts only on Rab7 or has also other targets.

a) I find it very critical that the authors analyze a Rab7 K191R mutant in their study to test if *retromer interaction* and *endosomal recycling* is restored.

b) This would also test if Rab7 is potentially hyperactive.

As suggested by the Reviewer, we analyzed the effect(s) of K191 ubiquitylation deficiency on Rab7 function. Because it is known that ubiquitylation can ‘hop over’ to a proximal lysine residue in the event that the preferred modification site is rendered unavailable [1], a double mutant K191R, K194R (described in the manuscript as 2KR) was used. Rab7-2KR exhibits reduced steady-state ubiquitylation levels as compared to wild type Rab7 (Supplementary Fig. 5c, d), which is consistent with a previous report demonstrating that K38 of Rab7 also serves as a ubiquitylation site [2].

1a) *In evaluating Rab7-2KR relative to wild type Rab7 we considered two possible*

outcomes with respect to recycling from the LE: either, disabling ubiquitylation leads to negation of the trafficking phenotype arising from depletion of USP32 or, alternatively, ubiquitylation deficient Rab7 fails to support cargo recycling from the LE. While the first possibility would indicate that ubiquitylation of Rab7 serves no purpose in the recycling process, the latter option would instead suggest that sequential addition and removal of this modification are needed to orchestrate Rab7-dependent cargo extraction from the late endosome.

Our observation that depletion of USP32 (and by extension ubiquitylation of Rab7) leads to accumulation of the cargo-selective retromer component VPS35 on the LE (Fig. 6d,e; formerly Fig. 5e,f) lends support to the second hypothesis above. Specifically, it suggests that in the absence of deubiquitylation by USP32, the retromer is 'stuck' on Rab7. This in turn implies that ubiquitylated Rab7 may attract VPS35. Consistent with this hypothesis, examination of Rab7-2KR revealed that, in cells depleted of endogenous Rab7, endosomes positive for GFP-Rab7-2KR exhibit less contact with structures harboring VPS35 as compared to wild type GFP-Rab7 under the same conditions (Fig. 6f,g).

Furthermore, we find that stable expression of GFP-Rab7-2KR in otherwise wild type cells (still harboring endogenous Rab7) is unable to rescue USP32 loss of function with respect to either LE enlargement or failure to resolve buds and tubules emanating from these disfigured LEs (Fig. 5; extended from former Fig. 4). At the same time, Rab7-2KR is still observed on the LE membrane (Figs. 4, 5) and successfully mediates transport of these organelles to the microtubule minus end (Fig. 4; for discussion see part **b** below), indicating that 2KR mutation in itself is not globally disruptive to the canonical Rab7 biology. These new findings support a model where both ubiquitylation and subsequent deubiquitylation of Rab7 promote a specialized role of this GTPase in LE recycling. If this ubiquitylation cannot be efficiently removed, as in the case of USP32 loss, not only is completion of this specialized function inhibited (as evidenced by failed fission of recycling tubules). In addition, in the absence of USP32, Rab7 accumulates on the LE membrane (Fig. 3g-i), resulting in less Rab7 available for ubiquitin-independent functions (see part **b** below).

1b) While comparing the behavior of LEs in cells stably expressing wild type GFP-Rab7 versus GFP-Rab7-2KR we noticed that the mutant cells displayed hyperclustering of LEs in the perinuclear region and exhibited slightly less motility than cells expressing GFP-Rab7 (Fig. 4a-c). This suggested that (C-terminal) ubiquitylation deficient Rab7 may promote minus-end-directed transport to a greater extent than its wild type counterpart, but would not subsequently recycle as efficiently. Indeed, in co-immunoprecipitation experiments considerably more Rab7-2KR was recovered on RILP than wild type Rab7 (Fig. 4d,e), and GFP-2KR alleviated late endosomal dispersion and immobility observed in GFP-Rab7-expressing cells under conditions of USP32 depletion (Fig. 4a-c).

Taken together with the results discussed in part **a** above, our findings suggest that ubiquitylation serves as a switch between canonical Rab7 functions, such as minus-end-directed LE transport, and its specialized role in recycling from the very same

organelles. In the event that the cycle of addition and removal of this modification is disrupted by loss of the cognate DUB USP32, a globally stalled late endosomal system results, characterized by swollen, aberrantly distributed and immobile endosomes. A model summarizing these observations on the effects of reversible ubiquitylation of Rab7 with respect to LE transport and cargo trafficking appears in Fig. 8.

2) *How do cells react to USP32 overexpression?* If I interpret the data right, then this has only been analyzed in Figure 3, but here also only with respect to ubiquitination (or better the loss of ubiquitination). I would expect that overexpression would mimic the Rab7 KR mutant.

While we can quantifiably detect reduced ubiquitylation of endogenous Rab7 in HeLa cells (Fig. 3c) and GFP-Rab7 in HEK293T cells (Fig. 3d) under conditions of ectopic USP32 expression, downstream phenotypes (i.e. endosome localization and motility) are less pronounced. We expect that this is likely due to the fact that endogenous levels of this DUB are already substantial, and therefore potent over-expression is challenging to achieve. Furthermore, it is not surprising that shifting the balance toward unmodified Rab7 (by overexpressing USP32) could reach the same levels of efficacy as complete inability to acquire such a modification in the first place (such as in the case of Rab7-2KR).

3a) The authors observe that endosomes are enlarged (Figure 4a) in the absence of USP32. Is the localization of Rab5 impaired under these conditions?

Yes, indeed, localization of Rab5-positive endosomes is perturbed in cells depleted of USP32. In fact, the distribution of Rab5 and Rab7 are essentially inverted under these conditions (Fig. 3e,f; Supplementary Fig. 4g), which is also reflected in the change in distribution of EEA1-positive versus CD63-positive compartments in response to loss of USP32 (Fig. 1b,d). Because minus-end-directed transport of early endosomes is coupled to their maturation [3-5], this suggests that Rab5-to-Rab7 transition may be perturbed when USP32 activity is inhibited. Unlike Rab7, Rab5 is not deubiquitylated by USP32 (Fig. 3b-d), and the observed effect on the Rab5-positive compartment likely occurs indirectly via Rab7. Because we find that membrane-associated fraction of Rab7 is elevated with USP32 depletion (Fig. 3g,h), we suspect that over time less Rab7 becomes available for newly internalized (Rab5-positive) endosomes in cells lacking USP32. As a consequence, Rab5 is not efficiently displaced by Rab7 and moves with these organelles towards the microtubule minus end.

3b) If Rab7 is localized to endosomes, is it still in the GTP-form and cannot bind retromer? This would suggest that only one of its functions is impaired (recycling), yet Rab7 may still contribute to fusion.

Rab7 still efficiently localizes to LE membranes in the absence of USP32, as evidenced by comparable overlap of Rab7 with the LAMP1 compartment (Fig. 3e; Supplementary Fig. 4f). On this basis it is expected that GTP loading of Rab7 is not affected by USP32. Additionally, membrane-associated fraction of Rab7 is actually increased in response to USP32 depletion (Fig. 3g,h). This suggests that elevated ubiquitylation of Rab7 (resulting from loss of USP32) essentially 'locks' Rab7 in place on its cognate membranes. Furthermore, because tubulation of Rab7-positive membranes is enhanced

in the absence of USP32, we hypothesize that ubiquitylated Rab7 exhibits improved association with the retromer complex. This supposition is supported by diminished overlap of GFP-Rab7-2KR with VPS35 as compared to that of wild type GFP-Rab7 in cells depleted of endogenous Rab7 (Fig. 6f,g). Collectively, these results suggest that ubiquitylation on Rab7 inhibit the second stage of its retromer-associated function, which requires deubiquitylation and likely concomitant removal of Rab7 from the affected membranes.

By contrast, RILP prefers the 2KR mutant to wild type Rab7 (Fig. 4d,e), and Rab7-2KR can rescue deficiency in minus-end-directed transport incurred by the absence of USP32, as evidenced by the restoration of the perinuclear LE compartment (Fig. 4a-c). Therefore, in this case ubiquitylated Rab7 inhibits transport (at least that mediated by RILP). Concerning the role of Rab7 in fusion, we find that, similar to RILP, the Rab7 effector PLEKHM1 also prefers Rab7-2KR mutant to wild type Rab7 (data not shown). However, we feel that inclusion of these data would further complicate the manuscript, and therefore fall outside the scope of this work.

Taken together, the key observations from new experiments performed following the initial submission support the model where the cycle of addition and removal of ubiquitin helps to switch Rab7 between its various functions at the LE membrane.

Reviewer #2 (Remarks to the Author):

Sapmaz et al., "USP32 mediates retrograde trafficking through deubiquitination of Rab7".

This is an interesting study with lots of nice data. It does however need revision before publication and set out below are the changes I think it needs. I have only one suggestion for additional experimental data, other changes are restricted to text and presentation of the figures.

We thank the Reviewer for the complimentary assessment of our study and the data presented therein. We also appreciate the Reviewer's constructive criticism and have addressed all of the points brought up as follows below.

1. The authors should determine if USP32 deubiquitination of Rab7 regulates the association of the retromer cargo-selective complex with Rab7. This can be achieved by expressing GFP-Rab7 and then immunoprecipitating GFP-Rab7 with its interacting proteins in cells treated with siRNA to knockdown USP32, or where USP32 is over expressed. Addition of this data will significantly strengthen the manuscript in my view.

The experiment suggested by the Reviewer has been attempted under various co-immunoprecipitation conditions; however, we found that retromer subunits could not be successfully recovered on either GFP-Rab7 or GFP-2KR. Inspection of other studies where such interactions were reported quickly revealed that these experiments are

typically performed using constitutively active GTP-locked Rab7 mutant Q67L [6, 7]. This approach would of course have been unsuitable for our purposes, as such a mutant would almost certainly override any contribution from ubiquitylation/deubiquitylation of Rab7. Therefore, to assess the effect of this reversible modification on Rab7 function in the context of the retromer complex, we analyzed the degree of contact between the cargo-selective retromer component VPS35 and either wild type Rab7 or Rab7-2KR in cells depleted of endogenous Rab7. These data appear in Fig. 6f,g and fit nicely with the accumulation of VPS35 on LEs affected by loss of USP32 (Fig. 6d,e). This is discussed further above under point 1a under comments of Reviewer 1.

2. I think the title should be, "USP32 regulates retrograde trafficking through deubiquitination of Rab7" as this better describes the role of USP32.

We thank the Reviewer for this suggestion. The title of the manuscript has been changed accordingly.

3. Some of the figures are poor. The micrographs are often overly busy with small images covered in lines etc. The position of the inset boxes sometimes does not correspond to the enlarged image. For example, the M6PR zooms in Fig 1 are showing a slightly different region on the original micrograph than indicated by the inset box. All of the figures with inset boxes need to be carefully checked to make sure that the zoomed image is correct. The presence of two M6PR zoom boxes in the diagram that is part of Fig 1 is confusing as it makes five zoom boxes but only four images are part of figure 1a. Overall, I was somewhat disappointed with the presentation of the data, it felt at times like a triumph of style over substance which is a shame as the data is generally good.

We appreciate Reviewer's constructive criticism of the data and have thoroughly revised data presentation throughout the manuscript. All figures have been adjusted to improve the clarity of data presentation.

Why does the lysosome in Figure 1 have a double membrane? Is it pretending to be an autophagosome?

There is no double membrane present on the endosomal structures of interest. The 'space' visible between the limiting MVB membrane and the surrounding cellular context is due to chemical fixation combined with sectioning of this aberrant organelle.

4. In addition to ref 24, the authors should also cite Rojas et al., (2008) (PMID 18981234) as evidence for Rab7 associating with and regulating the retromer complex.

The above reference has been cited [ref 26]. We regret our previous omission.

5. Line 228, "indespensible" is misspelled.

This has been corrected. We thank the Reviewer for pointing this out.

6. Line 247, I am at a loss to understand why the authors have cited refs 48,49 in a sentence describing the M6PR recycling back to the TGN via the retromer complex.

Indeed, previously selected references do not fit the associated discussion. Suitable references [17, 18 and 45] have now been selected instead. We apologize for the earlier mix-up.

7. In figure 5e, why is the MHC II used as a cargo for retrograde trafficking? Why not investigate the M6PR recycling?

Former Fig. 5e (presently Fig. 6d) is testing the association of retrograde trafficking machinery (VPS35) with the late endosomal compartment marked by MHC-II. M6PR trafficking is discussed in Fig. 7d and Supplementary Fig. 9a, b.

8. The inset boxes in Figure 6a don't correspond to the zooms very well.

This has been addressed (Supplementary Fig. 9a).

Reviewer #3 (Remarks to the Author):

The paper describes follow-up studies from an siRNA screen which identified the deubiquitinase (DUB) as a regulator of late endosome distribution in MelJuso studies. GFP-USP32 is mainly associated with the secretory pathway and concentrated in the TGN, yet a proteomic screen for potential USP32 substrates identified the late endosomal GTPase rab7 to be a potential substrate. USP32 depletion is also argued to disrupt recycling of ciM6PR from endosomes to Golgi, a process which is known to be rab7-dependent.

The overall results are certainly plausible but not entirely convincing at this stage. Ultimately there is no mechanism elucidated but a suggestion it may be through rab7 that is poorly supported. The paper is idiosyncratically written and the description of the state of the art vis a vis the endosomal system leaves much to be desired. Methods are lacking essential detail. There are also some experimental areas where quality could be improved. In my opinion not of sufficient general interest for Nature Comms.

*We thank the Reviewer for the constructive criticism and helpful questions and suggestions for improvement. We have addressed the majority of the issues raised. Questions pertaining to the role of reversible modification of Rab7 with ubiquitin have been dealt with in responses to **Reviewers 1 and 2** above. Additional specific points are discussed below.*

Ubiquitination of rab7 itself is already known from the cited paper (ref 31) and a further 45 papers identify the same Lys191 site according to phosphosite. No attempt is made to probe the function of this modification and it is not even clear if this is enriched on the

membrane associated fraction.

This is a fair point, which we have now addressed. In agreement with Reviewer's implicit expectation, membrane-associated fraction of Rab7 is indeed enriched in the absence of USP32 (Fig. 3g,h). This data supports the notion that unresolved ubiquitylation inhibits normal Rab7 cycling between its various functions on endosomes.

Nor do we ever see any endogenous rab7 blots.

An endogenous Rab7 blot appears in Fig. 6c; Supplementary Fig. 5a; Supplementary Fig. 8c. Additionally, endogenous Rab7 localization is examined as a function of USP32 depletion by immuno-fluorescence (Fig. 3e,f), and endogenous Rab7 is shown to colocalize with endogenous USP32 and VPS35 (Supplementary Fig. 8a).

Interaction with Vps35 has previously been shown by the Harper lab.

This is certainly the case and was cited accordingly in the original version of the manuscript.

Two other DUBs have been previously linked to M6PR recycling. Hao et al. claim USP7 is required for regulation of the WASH complex (Mol. Cell 2015), whilst Macdonald et al. show that depletion of USP8 also blocks M6PR recycling and that this may be linked to the stability of ESCRT-0. I was surprised that neither of these studies is referred to. Furthermore the authors need to show awareness of recent papers suggesting that ciM6PR retrieval is retromer independent.

We apologize for the earlier omission of relevant literature. This has been remedied by the inclusion of references (47 and 67). In addition, we now include USP8 depletion as a control for LE-to-TGN transport dysfunction (Movie S1a, b). We apologize for the earlier omission of recent papers suggesting that CI-M6PR retrieval is retromer independent. This has been remedied by the inclusion of references (52 and 53)

Specific Comments:

1. Details/data of the siRNA screen are lacking. Was the original screen performed with pooled oligos? how many hits? Do all oligos used in the original screen deconvolute.

As per Reviewer's request, the original siRNA screen targeting human DUBs in MelJuSo cells has now been included in the manuscript (Fig. 1a). The corresponding experimental details can be found in the Materials and Methods section under the heading 'siRNA-based screen for DUBs'.

The initial screen was indeed performed using pooled oligos in MelJuSo cells according to a previously described methodology [8] using cell surface levels of late endosomal cargo MHC class II (MHC-II) as a read-out. Those DUBs whose depletion was found to perturb MHC-II surface levels at or above the chosen threshold z-score threshold of +/-3

are shown in Fig. 1a. Because of our interest in late endosomal architecture and dynamics (as stated in the introduction), the original DUB 'hits' were subsequently analyzed for intracellular MHC-II phenotypes by microscopy prior to deconvolution of siRNA pools. Intracellular phenotypes were scored on two criteria: size and distribution of vesicles carrying MHC-II. Those DUB hits, which showed vesicle enlargement and/or redistribution were followed up on with deconvolution. DUBs whose depletion-associated phenotypes deconvolved with at least 2 or more independent oligos are shown in Supplementary Fig. 1. These details have now been added to the relevant Methods and Materials sections of the manuscript.

2. Fig. 1- how is dispersion quantitated? error bars and statistical tests are not defined throughout. Is this the best way to quantitate in part C- what about some visual representation of pixels.

Pixel analyses termed 'Fractional Distances' have now been added throughout the manuscript to report on intracellular vesicle distribution. This type of unbiased quantification of pixel spread was performed according to a previously published method [9]. The first instance of fractional distance quantification in the revised manuscript appears in Fig. 1d. We thank the Reviewer for the suggestion to improve this.

3. Fig 2. I could not find details of the anti-USP32 antibody. What is it? How was it validated for IF and em? In other words need to show siRNA controls.

The anti-USP32 antibody details can be found in the Materials and Methods section corresponding to antibodies and fluorescent dyes. This monoclonal antibody raised in mouse was validated using depletion (and rescue) of USP32 for western blot (Figs. 1c, 2e) and immunofluorescence (Fig. 2a and Supplementary Fig. 2). In all of the above applications the antibody performed well for analysis of both endogenous and ectopically expressed USP32.

Fig 2d. the re-expressed USP32 does not look like the same distribution as the endogenous-is this correct. I could not find details of the siRNA resistant construct.

This is actually not correct. Distribution of ectopically expressed USP32 closely mimics that of the endogenous enzyme. Both populate primarily the Golgi/TGN in MeJuSo (Fig. 2a,d; Supplementary Fig. 2a) as well as HeLa (Supplementary Fig. 2a) cells.

4. Figure 3. Schematic needs to highlight these experiments use different cell types. Rab7A is not the most sensitive protein to USP32 depletion- what are these other proteins? There should be a spreadsheet for these experiments so readers can look up individual sites. Raw data should be made available through repositories. To my mind this experiment should be done with both oligos.

d. need to be more careful and consistent with labelling they are looking at GFP-RAB7A in all cases. Why not look at endogenous rab7.

In-gel ubiquitylation assays, particularly those probing changes in a mono- or short-chain-ubiquitylation event, are often challenging to perform on endogenous material. This is typically due to difficulties associated with detection of one or two ubiquitin molecules present on a modified target (in contrast to poly-ubiquitylation, which is much

easier to detect). For this reason HA-tagged ubiquitin is commonly used in this type of assay. Furthermore, because under normal conditions only a fraction of the total cellular target (in this case Rab7) is modified at any given time, ectopic expression of both the target and HA-ubiquitin in the same cells allows robust and therefore quantifiable detection of this modification under different treatment conditions. The fact that the modification in question is relevant to endogenous Rab7 is attested to by its very identification in different cell lines under different biological conditions (Fig. 3a-c): +/- USP32 depletion (performed in MeJuSo cells) and +/- USP32 overexpression (performed in HeLa cells).

5. Figure 5. Endogenous staining for USP32 in HeLa cells does not look like the TGN- how does it compare with epitope tagged? panel e looks over-exposed. Error bars and stats?

USP32 staining in HeLa cells colocalizes with TGN markers in both MeJuSo and HeLa cell lines (Fig. 2a, Supplementary Fig. 2). Also see the last part under point 3 above. The data referred to in former Fig. 5e has been revised (presently Fig. 6d).

6. Figure 6 legend EGFR phosphosite needs to be specified.

panel g/h. The data showing a major block to EGFR degradation contradict the findings of Savio et al. Curr Biol (2015)- need to comment on this and also show with a second oligo.

A generic anti-phosphotyrosine antibody (mouse monoclonal 4G10) was used to detect overall levels of activated EGFR. This information now appears in the legend corresponding to Fig. 7f. The block in ligand-mediated degradation of EGFR has now been recapitulated with 2 additional siRNAs targeting USP32. The combined quantification appears in Fig. 7g, and additional western blots can be found in the Supplementary Fig. 9c. In our opinion the data on this point is sufficiently conclusive and fits with both defects observed as a consequence of persistent Rab7 ubiquitylation: namely inhibition of minus-end-directed transport of late endosomes (Fig. 4) and inhibition of recycling from these compartments to the TGN (Figs. 5 and 6).

7. They should ideally measure cathepsin D secretion into the medium.

No cathepsin D secretion was measured under conditions of USP32 depletion.

However, cleavage of mature cathepsin D was observed in cells lacking the DUB (data not shown).

References

1. Kim, B.J., et al., *The Histone Variant MacroH2A1 Is a BRCA1 Ubiquitin Ligase Substrate*. Cell Rep, 2017. **19**(9): p. 1758-1766.
2. Song, P., et al., *Parkin Modulates Endosomal Organization and Function of the Endo-Lysosomal Pathway*. J Neurosci, 2016. **36**(8): p. 2425-37.
3. Nielsen, E., et al., *Rab5 regulates motility of early endosomes on microtubules*. Nat Cell Biol, 1999. **1**(6): p. 376-82.
4. Rink, J., et al., *Rab conversion as a mechanism of progression from early to late endosomes*. Cell, 2005. **122**(5): p. 735-49.
5. Poteryaev, D., et al., *Identification of the switch in early-to-late endosome transition*. Cell, 2010. **141**(3): p. 497-508.
6. Rojas, R., et al., *Regulation of retromer recruitment to endosomes by sequential action of Rab5 and Rab7*. J Cell Biol, 2008. **183**(3): p. 513-26.
7. Purushothaman, L.K., et al., *Retromer-driven membrane tubulation separates endosomal recycling from Rab7/Ypt7-dependent fusion*. Mol Biol Cell, 2017. **28**(6): p. 783-791.
8. Paul, P., et al., *A Genome-wide multidimensional RNAi screen reveals pathways controlling MHC class II antigen presentation*. Cell, 2011. **145**(2): p. 268-83.
9. Jongsma, M.L., et al., *An ER-Associated Pathway Defines Endosomal Architecture for Controlled Cargo Transport*. Cell, 2016. **166**(1): p. 152-66.

Reviewers' comments:

Reviewer #1 (Remarks to the Author):

The authors provide a very extensive revision, in which they now include the KR mutant of Rab7 to recapitulate some of the findings observed by depleting USP32. Their data agree with a model that ubiquitination modifies the role of Rab7 to function in retromer-mediated recycling vs. Rab7-dependent transport (i.e. interaction with PLEKHM1). The authors demonstrate this also by pull-down with RILP with Rab7 (wt and mutant). To complete this figure, I feel that one additional control would be very informative. The authors should include the siUSP32 and control, which should likewise affect the interaction, thus showing that both mutant and absence of ubiquitination show the same effect. They may have done this already.

Apart from this, I have no further requests.

--

Reviewer #2 (Remarks to the Author):

The authors have made revisions to the manuscript based on the concerns I raised previously although I note that the most important experiment, does USP32 regulate Rab7a interaction with retromer? has not been done - or there is no data indicating it has been done. The authors claim that this is because retromer cannot be recovered from pulldowns of Rab7a unless the Q67L (active) mutant is used. This is not true. In the study by Seaman et al., (2009) (see PMID: 19531583), it was shown that GFP-Rab7a (wildtype) could pulldown sufficient retromer for bands to be visible on silver stained gels. This is a key experiment and I do not think the study should be published without this data.

--

Reviewer #3 (Remarks to the Author):

Fig. 3g,h- does not address the question as to whether the ubiquitinated form of rab7 is enriched on membranes.

It is still very difficult to get a handle on how well their original screen deconvolved- I really want to know how many oligos targeting USP32 reproduced this behaviour and how many did not and if this correlates with efficiency of depletion.

No attempt to address the substantive part of my point 4. Still no obvious means of accessing the full ms dataset.

Savio reference is included but discrepancies with the current data are not addressed.

Error bars- SD on n=2.

Below you can find a detailed point-by-point discussion of the issues raised in the last round of review of our manuscript entitled “USP32 regulates late endosomal transport and recycling through deubiquitylation of Rab7”. We have made substantial revisions to the manuscript figures and text, as well as updated the Supplementary materials to reflect all remaining Reviewer concerns. We thank the Reviewers for their constructive criticisms and continued support of our study. We feel the new changes have further strengthened the work and hope the Reviewers find it ready for publication in Nature Communications.

Reviewer #1 (Remarks to the Author):

The authors provide a very extensive revision, in which they now include the KR mutant of Rab7 to recapitulate some of the findings observed by depleting USP32. Their data agree with a model that ubiquitination modifies the role of Rab7 to function in retromer-mediated recycling vs. Rab7-dependent transport (i.e. interaction with PLEKHM1). The authors demonstrate this also by pull-down with RILP with Rab7 (wt and mutant). To complete this figure, I feel that one additional control would be very informative. The authors should include the siUSP32 and control, which should likewise affect the interaction, thus showing that both mutant and absence of ubiquitination show the same effect. They may have done this already.

Apart from this, I have no further requests.

We thank the Reviewer for the complimentary assessment of our previous revision efforts, including the addition of a new mechanistic angle to the story via the Rab7-RILP axis and incorporation of a new tool to dissect relevant mechanistic details—the Rab7-2KR mutant.

The new experiment requested by the Reviewer to directly probe the effect of USP32 depletion on the interaction between Rab7 and RILP has been successfully performed. The new results can be found in Fig. 6f and Supplementary Fig. 7d along with the corresponding text in the Results section (lines 227-231). As expected, we find that loss of USP32 exhibits a negative effect on the association of RILP with wild type Rab7, suggesting that ubiquitylated Rab7 does not bind RILP as efficiently as its unmodified counterpart. Although modest, the effect is consistent with the preference displayed by RILP for Rab7-2KR—a mutant lacking the capacity for ubiquitylation on its C-terminal K191 (targeted by USP32) or the neighboring K194 as compared to wild type Rab7 (Fig. 6d, e). The new data further substantiate our model (Fig. 9), where we propose that ubiquitylation of Rab7 helps to switch this multifaceted GTPase from dynein-mediated vesicle transport towards another function, namely recycling from the LE/MVB.

We thank the Reviewer for this insightful experimental suggestion, which has helped strengthen our conclusions and make the manuscript more complete.

Reviewer #2 (Remarks to the Author):

The authors have made revisions to the manuscript based on the concerns I raised previously although I note that the most important experiment, does USP32 regulate Rab7a interaction with retromer? has not been done - or there is no data indicating it has been done. The authors claim that this is because retromer cannot be recovered from pulldowns of Rab7a unless the Q67L (active) mutant is used. This is not true. In the study by Seaman et al., (2009) (see PMID: 19531583), it was shown that GFP-Rab7a (wildtype) could pulldown sufficient retromer for bands to be visible on silver stained gels. This is a key experiment and I do not think the study should be published without this data.

We agree with the Reviewer that testing the effect of USP32 on the interaction between Rab7 and the retromer would substantially strengthen the manuscript. Due to the fact that reliable co-IP between GFP-Rab7 and either endogenous or ectopically expressed VPS35 could not be consistently detected in our hands, as examined independently by AS and IB in HeLa, HEK293, and MeJuSo cells under either our standard co-IP conditions (described in the Materials and Methods section) or those reported by Seaman et al., 2009, (including treatment of cells with nocodazole prior to IP), we chose a different strategy to monitor this interaction, as described below.

In order to probe Rab7 interactions occurring inside living cells (i.e. prior to lysis) as a function of its ubiquitylation status, we introduced a promiscuous biotin ligase domain [1] between the GFP tag and either wild type or mutant Rab7 sequence, yielding GFP-BirA-Rab7 and GFP-BirA-2KR, respectively. Following transfection of these constructs and incubation of cells with membrane-permeable biotin, biotinylation of specific proteins of interest could be assayed by Western blot analysis of neutravidin precipitates. The results of these efforts can now be found in Fig. 7f-h and Supplementary Fig. 8f, along with the corresponding text in the Results (lines 250-260). The described experiments offer further support to our model (Fig. 9), as briefly summarized below.

Upon addition of biotin to cells expressing GFP-BirA-Rab7, versus those transfected with free GFP-BirA fusion as a negative control, biotinylation of endogenous (Supplementary Fig. 8f) and ectopically expressed VPS35 was observed (Fig. 7f). Labeling of VPS35 with biotin diminished in the presence of GFP-BirA-2KR as compared to wild type GFP-BirA-Rab7 (Fig. 7f, g), suggesting that lack of (C-terminal) ubiquitylation on Rab7 negatively affects its interaction with the retromer. Conversely, depletion of USP32 resulted in elevated biotinylation of VPS35 compared to control (siCtrl) (Fig. 7h; Supplementary Fig. 8f). In the same experiment, labeling of endogenous USP32 (siCtrl condition) was observed in the presence of GFP-BirA-Rab7, indicating that USP32 and Rab7 are in the same complex. These new results are consistent with the observed increase in colocalization between MHC class II and VPS35 in cells depleted of USP32 (Fig. 7d, e), as well as with diminished colocalization of GFP-2KR with VPS35 as compared to wild type GFP-Rab7 (Supplementary Fig. 8c, d). Taken together with perturbations in tubule resolution from the Rab7-positive endosome observed in the absence of USP32 activity (Fig. 8), the results described above suggest that both ubiquitylation and de-ubiquitylation of Rab7 benefit membrane recycling from the Rab7 compartment.

On the basis of our observations, we propose a model whereby USP32 regulates LE/MVB lifestyle in two ways (Fig. 9). Firstly, deubiquitylation of Rab7 by USP32 promotes transport of Rab7-positive vesicles, such as by making Rab7 available to its effector RILP. Secondly, USP32 promotes fission of recycling tubules emanating from the same endosomes, as well as helps to release Rab7 from the endosomal membrane for subsequent functional cycles.

We thank the Reviewer for his/her constructive criticism, encouraging us to further substantiate our initial findings and proposed model towards a better-rounded manuscript.

Reviewer #3 (Remarks to the Author):

It is still very difficult to get a handle on how well their original screen deconvolved- I really want to know how many oligos targeting USP32 reproduced this behaviour and how many did not and if this correlates with efficiency of depletion.

To address this point, we have now included USP32 siRNA pool deconvolution data in the Supplementary Fig. 1a and b, showing that all 4 oligos (labeled as P1-P4) efficiently downregulate USP32 protein abundance and show significant reduction in surface levels of MHC class II, as measured by flow cytometry. In addition, we found that one of the two previously reported siRNA sequences against USP32 [2] (labeled as oligo #2 in the Supplementary Fig. 1) performed best with regards to endosomal defects as well as provided excellent depletion (Supplementary Fig. 1a, c). We therefore proceeded primarily with oligo #2 throughout the manuscript, unless otherwise indicated. This information is now clarified in the Materials and Methods. Importantly, the rescue experiment reported in Fig. 3 was performed using oligo #2, unequivocally demonstrating that both vesicular perturbations observed with USP32 depletion (i.e. mislocalization and swelling) are dependent on the catalytic activity of this enzyme.

To better clarify the screening workflow and our selection of USP32 as the DUB of interest for the study, we have now extensively revised the Results section corresponding to Fig. 1.

No attempt to address the substantive part of my point 4. Still no obvious means of accessing the full ms dataset.

All the MS data referred to in the manuscript have been deposited to the ProteomeXchange Consortium via the PRIDE partner repository with the dataset identifier 1-20181109-74057.

Fig. 3g,h- does not address the question as to whether the ubiquitinated form of rab7 is enriched on membranes.

While we agree that the experiment referred to by the Reviewer (now presented in Fig. 5g, h) does not directly address the cellular residence of ubiquitinated Rab7, it does unequivocally show that under conditions of USP32 depletion, which directly leads to more Rab7 ubiquitination, Rab7 accumulates on membranes. These data therefore suggest that ubiquitinated Rab7 (of the type arising in response to lack of deubiquitination by USP32) resides preferentially on the membrane, and in the manuscript text we specifically use the word 'suggest' to reflect this. Also, ubiquitylated Rab7 has been removed from the model in Fig. 5i.

Savio reference is included but discrepancies with the current data are not addressed.

Our understanding is that the Reviewer is asking us to explain how USP32 can be a positive regulator of EGFR turnover despite the fact that Savio *et. al* did not identify USP32 as such in their screen. Firstly, we would like to point out that we initially picked up USP32 as a modulator of MHC class II abundance at the cell surface, not as a regulator of EGFR turnover, using siGENOME pool of 4 siRNAs targeting USP32 from Dharmacon. By contrast, Savio *et. al* used 2 different oligos targeting this protein in their study, where screening was performed in another model system. While we cannot speak for the pros and cons of the oligos and validation techniques Savio *et. al* used to interrogate USP32 activity in their system versus ours, we find that in addition to affecting MHC class II traffic, loss of USP32 also dampens the rate of EGFR turnover. This phenotype is observed with at least 3 independent siRNAs targeting USP32—oligos #2, #P1 and #P4, all of which give efficient depletion of endogenous USP32 protein in HeLa cells (Fig. 2f, g and Supplementary Fig. 2).

Error bars- SD on n=2.

Because it is unclear which experiment is referred to here, we give a full account of error quantification throughout the manuscript below.

Identification of the hits in the DUB screen for altered MHC-II surface levels is based on 3 independent measurements. The screen was performed twice; the first experiment included biological duplicates (n1 and n2), where independent cell culturing, transfections and immunostainings were performed in parallel by RHW and IB. The second experiment was performed as a singlet by RHW (n3). The previous version of the manuscript mistakenly stated that the screen was performed as n=2, and this has now been corrected in the legend corresponding to Fig. 1a. We apologize for this error.

Throughout the manuscript SD on n=2 has been used for microscopy experiments on fixed samples, where overlap coefficients are reported. In these cases, >10 cells per condition per experiment were typically used, as is now indicated in every legend, and data derived from all individual images were used to calculate the error. Because multiple cells often appear in the same image (data was collected using 63x magnification in combination of 2.0x digital zoom), reporting of averages and errors

typically relied on a total of 7-10 values (or more) for the two experiments combined. In some experiments reporting colocalization, $n=3$ was used, as indicated, but the same methodology to error reporting for image quantification was applied.

For fractional distance quantification, pixel spread of vesicle staining from at least 10 cells per condition per experiment ($n=2$) was analyzed as described in Jongsma and Berlin et. al, 2016. This resulted in many data points analyzed, all of which are plotted in the corresponding graphs.

Proteomics data analysis was performed as described in detail in the Material and Methods section under 'Ubiquitome analysis'. The SILAC experiment was performed once for both siUSP32/siCtrl and USP32-HA/Ctrl overexpression. The results of the former setup were independently validated in a label-free LFQ experiment performed with 2 technical replicates. The latter was independently corroborated by experiments reported in Fig. 4d and quantified in Supplementary Fig. 5e ($n=3$ for Rab7).

Quantification of all gel-based experiments is reported on the basis of at least 3 independent values ($n=3$), as indicated in the corresponding legends.

We thank the Reviewer for his/her detailed points and suggestions to clarify our methods and improve the manuscript.

References

1. Roux, K.J., et al., *A promiscuous biotin ligase fusion protein identifies proximal and interacting proteins in mammalian cells*. J Cell Biol, 2012. **196**(6): p. 801-10.
2. King, R.W., et al., *Mitotic progression genes and methods of modulating mitosis* 2008, Sanofi-Aventis France, Harvard College

REVIEWERS' COMMENTS:

Reviewer #1 (Remarks to the Author):

The authors addressed my request, and observed a very minor effect upon depletion of USP32 on the interaction between RILP and Rab7. Maybe this is as much as can be expected with such a complex system.

I just have one minor last issue. The authors show in Figure 9 the release of Rab7. To my knowledge, Rab7 will be accompanied by GDI if released from membranes. This should be added.

Reviewer #2 (Remarks to the Author):

The experiment I suggested, a pulldown of GFP-Rab7a has not been done but the authors have used the BirA system instead. To me, this seems like taking a sledge-hammer to crack a nut but the goal of experiment - to evaluate the interaction between Rab7a and retromer - has been achieved. In my view, the manuscript is now suitable for publication.

Dear Editors,

Below you can find a point-by-point response to the remaining comments provided by the Referees with regards to the resubmission of our manuscript NCOMMS-17-21846B entitled “USP32 regulates late endosomal transport and recycling through deubiquitylation of Rab7”. We extend our appreciation for the constructive input of the Reviewers throughout the peer review process and thank them for helping us improve the content and presentation of our manuscript for publication in *Nature Communications*.

REVIEWERS' COMMENTS:

Reviewer #1 (Remarks to the Author):

The authors addressed my request, and observed a very minor effect upon depletion of USP32 on the interaction between RILP and Rab7. Maybe this is as much as can be expected with such a complex system.

As stated in our original rebuttal letter, we agree with the Reviewer that the overall effect on the interaction between RILP and Rab7 under condition of USP32 depletion in combination with over-expression of the interacting components is modest. However, keeping in mind that the total proportion of ubiquitylated GFP-Rab7 at any given time under control conditions is not likely to rise above 10-20% (as can be approximated from anti-GFP immunoblots in ubiquitylation assays), the effect we observe (15-20% decrease in Rab7/RILP co-IP) is actually rather profound.

We thank the Reviewer for suggesting we perform this experiment and for the favorable assessment of our manuscript for publication.

I just have one minor last issue. The authors show in Figure 9 the release of Rab7. To my knowledge, Rab7 will be accompanied by GDI if released from membranes. This should be added.

This information has now been incorporated into Fig. 9 and the corresponding legend, reflecting Reviewer's suggestion.

Reviewer #2 (Remarks to the Author):

The experiment I suggested, a pulldown of GFP-Rab7a has not been done but the authors have used the BirA system instead. To me, this seems like taking a

sledge-hammer to crack a nut but the goal of experiment – to evaluate the interaction between Rab7a and retromer – has been achieved. In my view, the manuscript is now suitable for publication.

We thank the Reviewer for the recognition of the effort it took our team (now nicknamed ‘the sledgehammer’) to address his/her experimental request. We fully agree that performing this experiment added substantially to the molecular aspect of our study and appreciate the Reviewer’s positive assessment of our manuscript as suitable for publication.